# On Bias-Variance Alignment in Deep Models

**Lin Chen, Michal Lukasik, Wittawat Jitkrittum, Chong You, Sanjiv Kumar**
Google Research, {linche,mlukasik,wittawat,cyou,sanjivk}@google.com

## Abstract

Classical wisdom in machine learning holds that the generalization error can be decomposed into bias and variance, and these two terms exhibit a *trade-off*. However, in this paper, we show that for an ensemble of deep learning based classification models, bias and variance are *aligned* at a sample level, where squared bias is approximately *equal* to variance for correctly classified sample points. We present empirical evidence confirming this phenomenon in a variety of deep learning models and datasets. Moreover, we study this phenomenon from two theoretical perspectives: calibration and neural collapse. We first show theoretically that under the assumption that the models are well calibrated, we can observe the bias-variance alignment. Second, starting from the picture provided by the neural collapse theory, we show an approximate correlation between bias and variance.

## 1 Introduction

The concepts of *bias* and *variance*, obtained from decomposing the generalization error, are of fundamental importance in machine learning. Classical wisdom suggests that there is a trade-off between bias and variance: models of low capacity have high bias and low variance, while models of high capacity have low bias and high variance. This understanding served as an important guiding principle for developing generalizable machine learning models, suggesting that they should be neither too large nor too small (Bishop, 2006). Recently, a line of research found that deep models defy this classical wisdom (Belkin et al., 2019): their variance curves exhibit a unimodal shape that first increases with model size, then decreases beyond the point that the models can perfectly fit the training data (Neal et al., 2018; Yang et al., 2020). While the unimodal variance curve explains why over-parameterized deep models generalize well, there is a lack of understanding on why it occurs.

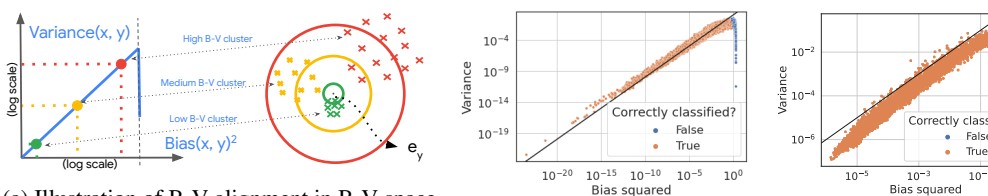

(a) Illustration of B-V alignment in B-V space (left) and output space (right).

(b) B-V for a vision task

(c) B-V for an NLP task

Figure 1: The bias-variance alignment phenomenon. *(a)* Given an input $x$ and its associated label $y$, bias-variance alignment refers to the phenomenon that the bias and variance of a deep model satisfy $\log \text{Bias}^2_{h_\theta,(x,y)} \approx \log \text{Vari}_{h_\theta,(x,y)}$ for correctly classified points, as illustrated as the dashed line (see the left subfigure). In the right subfigure, each cross represents a prediction of the model ensemble $\{h_\theta\}$ on a sample $x$, and the center is the one-hot encoding $e_y$ of the corresponding label $y$. Green, yellow, and red colored clusters of crosses correspond to the three groups also shown in the left subplot, with small, medium, and large bias, respectively. Bias-variance alignment implies that the three groups have small, medium, and large variance, respectively. *(b)* Per-example bias and variance for ResNet-50 trained on ImageNet, where each dot corresponds to a test sample and colored according to whether the sample is correctly classified by the model or not. Bias and variance are estimated from 20 independently trained networks with different initial weights and over different bootstrap samples from the train set, following methodology from Neal et al. (2018). *(c)* Per-example bias-variance for BERT trained on TREC data (see Section E.7 in Appendix for details).

This paper revisits the study of bias and variance to understand their behavior in deep models. We perform a *per-sample* measurement of bias and variance in popular deep classification models. Our study reveals a curious phenomenon, which is radically different from the classical tradeoff perspective on bias-variance, while is concordant with more recent works (Belkin et al., 2019; Hastie et al., 2022; Mei & Montanari, 2022). Given a sample $x$ and its corresponding label $y$ from a dataset of test examples $\{(x_i, y_i)\}_{i \in [n]}$, let $\text{Bias}_{h_\theta, (x,y)}$ and $\text{Vari}_{h_\theta, (x,y)}$ be the bias and variance, respectively, of an ensemble of deep models $\{h_\theta\}$. Here, the randomness in calculating the bias and variance comes from $\theta$, which depends on the randomness in parameter initialization, batching and sampling of the training data (Neal et al., 2018; Yang et al., 2020).

Our key empirical observations can be summarized with the following two statements which we call the *Bias-Variance Alignment* and the *Upper Bounded Variance*. As we explain in the remainder of the paper, and as summarized in Table 1, these observations encapsulate our empirical observations, and also capture the special cases we prove from the calibration and the neural collapse assumptions.

**Bias-Variance Alignment.** First, we find that for correctly classified points $(x, y) \in \{(x_i, y_i)\}_{i \in [n]}$:

$$\log \text{Vari}_{h_\theta, (x_i, y_i)} \approx \log \text{Bias}^2_{h_\theta, (x_i, y_i)} + E_{h_\theta}, \qquad (E_{h_\theta} \text{ is a constant independent of } i)$$

$$\text{or} \quad \log \text{Vari}_{h_\theta, (x_i, y_i)} = \log \text{Bias}^2_{h_\theta, (x_i, y_i)} + E_{h_\theta} + \varepsilon_i, \quad (\varepsilon_i \text{ is noise s.t. } \mathbb{E}_{i \sim \text{Unif}([n])}[\varepsilon_i] = 0) \tag{1}$$

where $\varepsilon_i$ is random noise with mean vanishing across the dataset (i.e., $\mathbb{E}_{i \sim \text{Unif}([n])}[\varepsilon_i] = 0$). Specifically, our quantitative results show that (1) a simple linear regression of $\log \text{Vari}_{h_\theta, (x_i, y_i)}$ on $\log \text{Bias}^2_{h_\theta, (x_i, y_i)}$ yields a remarkably high coefficient of determination $R^2$; (2) the residuals of the simple linear regression exhibit an approximate normal distribution (we provide evidence for (1) and (2) in Section 3.1).

| Type | Assumptions | Finding (logarithmic scale) | Finding (linear scale) | Ref. | Note |
|---|---|---|---|---|---|
| Empirical | Large model size + Correctly classified data | $\log \text{Vari}_{h_\theta, (x_i, y_i)}$ $\approx \log \text{Bias}^2_{h_\theta, (x_i, y_i)} + E_{h_\theta}$ | $\text{Vari}_{h_\theta, (x_i, y_i)}$ $= C_{h_\theta} \text{Bias}^2_{h_\theta, (x_i, y_i)} + \xi_i$ | Sec. 3 | |
| Theoretical | Perfect calibration | | $\text{Bias}^2_{h_\theta, (x_i, y_i)} \approx \text{Vari}_{h_\theta, (x_i, y_i)}$ | Sec. 4 | a |
| Theoretical | Neural collapse + Binary classification | $\frac{\log \text{Bias}^2_{h_\theta, (x_i, y_i)}(k)}{\log \text{Vari}_{h_\theta, (x_i, y_i)}(k)} \in (1.114, 4)$ | $\frac{\text{Bias}^2_{h_\theta, (x_i, y_i)}}{\text{Vari}_{h_\theta, (x_i, y_i)}} \in \left( \frac{(2s-1)^2}{\exp(2s)}, 3 \right)$ | Sec. 5 | b |

ᵃ The result $\text{Bias}^2_{h_\theta, (x,y)} \approx \text{Vari}_{h_\theta, (x,y)}$ corresponds to $C = 1$ in our main empirical observation on the linear scale presented in Eq. (2) and we bound $\xi_i = \text{Vari}_{h_\theta, (x,y)} - \text{Bias}^2_{h_\theta, (x,y)}$ by the calibration error in Section 4.
ᵇ Here, $k \in \{1, 2\}$ is class index and $s$ is (roughly speaking) the $\ell_2$ norm of the prelogit of $h_\theta(x)$ (i.e., before softmax).

Table 1: A summary of our findings on the bias-variance alignment.

In linear scale, we can represent Equation (1) as

$$\text{Vari}_{h_\theta, (x_i, y_i)} = C_{h_\theta} \text{Bias}^2_{h_\theta, (x_i, y_i)} + \xi_i, \quad (C_{h_\theta} = e^{E_{h_\theta}} \mathbb{E}_{i \sim \text{Unif}([n])}[e^{\varepsilon_i}] > 0 \text{ is a constant})$$

$$\xi_i = O(\text{Bias}^2_{h_\theta, (x_i, y_i)})\eta_i, \qquad (\eta_i \text{ is noise s.t. } \mathbb{E}_{i \sim \text{Unif}([n])}[\eta_i] = 0) \tag{2}$$

A formal statement of the above formulations can be found in Proposition E.1 in Appendix E.4. Note that because the noise term $\xi_i$ scales with squared bias, Equation (2) predicts that the sample-wise bias-variance in linear scale has a cone-shaped distribution (i.e., as bias increases, an increasingly wider range of variance is covered by examples). We discuss this in more detail in Appendix E.4.

**Upper Bounded Variance.** Second, we find that the following relation approximately holds for all examples (i.e., for both correctly and incorrectly classified examples):

$$\text{Bias}^2_{h_\theta, (x,y)} \geq C_{h_\theta} \cdot \text{Vari}_{h_\theta, (x,y)}, \quad \forall (x, y) \in \{(x_i, y_i)\}_{i \in [n]}, \tag{3}$$

where $C_{h_\theta}$ is the same constant as in Equation (2). In Figure 1, we illustrate these findings on an illustrative example. Observe that for correctly classified sample points, the bias and variance align closely along the line of $\text{Bias}^2_{h_\theta, (x,y)} = \text{Vari}_{h_\theta, (x,y)}$, i.e., the equality in (1) holds. We refer to this phenomenon as the *bias-variance alignment*. For incorrectly classified samples, we observe $\text{Bias}^2_{h_\theta, (x,y)} > \text{Vari}_{h_\theta, (x,y)}$, hence the inequality in (3) approximately holds for all examples. It is worth noting that Eq. (3) provides an explanation for why deep models have limited variance, and in

effect, good generalization, i.e., the variance of a model is always bounded from above by the squared bias at every sample.

The paper provides both empirical and theoretical analyses of the bias-variance alignment phenomenon. We organize the paper as follows. We begin with empirical investigations, in which we observe the bias-variance alignment across architectures and datasets (Section 3). We then move on to theoretical explanations of these phenomena. We start from a statistical perspective, where we connect calibration and the bias-variance alignment (Section 4). In the process, we generalize the theory from previous works on calibration implying the generalization-disagreement equality (Jiang et al., 2022; Kirsch & Gal, 2022) (Section 4.1). Next, we show how starting from a separate perspective of neural collapse (Papyan et al., 2020) can lead to the bias-variance approximate equality result (Section 5). We conclude with the discussion of wider implications of our findings (Section 6).

Our main contributions are: (1) We conduct experiments to show that the bias-variance alignment holds for a variety of model architectures and on different datasets. (2) We provide evidence that the phenomenon does not occur if the model is small. This suggests that the bias-variance alignment is specific to large neural networks and provides more evidence that there could be a sharp difference between small and large models. (3) Theoretically, we prove the bias-variance alignment under the assumption that the model is well-calibrated (i.e., the output of the softmax layer aligns with the true conditional probability of each class given the data). As a side product, we provide a unified definition for a variety of definitions of calibration introduced in previous works. (4) We show that the neural collapse theory predicts the approximate bias-variance alignment.

## 2 BACKGROUND AND RELATED WORK

### 2.1 BACKGROUND ON BIAS-VARIANCE DECOMPOSITION

Consider the task of learning a multi-class classification model $h_\theta : \mathcal{X} \to \mathcal{M}([K]) \subseteq \mathbb{R}^K$, where $\mathcal{X}$ is the input domain, $\mathcal{M}([K])$ is the set of distributions on $[K]$, and $K$ is the number of classes. Let $\{h_\theta : \mathcal{X} \to \mathcal{M}([K])\}$ be an ensemble of trained models, where $\theta$ is a random variable taking values from $\Theta$[1]. For any input $x \in \mathcal{X}$, we use $h_\theta(\cdot \mid x)$ to represent the corresponding distribution. That is, $h_\theta(\cdot \mid x) \triangleq (h_\theta(1 \mid x), \ldots, h_\theta(K \mid x))$ is the vector of predictive probabilities from model $h_\theta$. Given any sample $(X, Y) \in \mathcal{X} \times [K]$, the bias and variance of $\{h_\theta\}$ with respect to the mean squared error (MSE) loss are defined as follows.

**Definition 2.1** (Bias and Variance). Let $h(\cdot \mid x) \triangleq \mathbb{E}_\theta h_\theta(\cdot \mid x)$ be the mean function of $\{h_\theta\}$. The bias, variance, and bias-variance gap of the $i$-th entry on $(X, Y)$, for each $i \in [K]$, are defined as

$$\text{Bias}_{h_\theta,(X,Y)}(i) = \beta_{h_\theta,(X,Y)}(i) \triangleq |h(i \mid X) - \mathbf{1}\{Y = i\}| , \tag{4}$$

$$\text{Vari}_{h_\theta,(X,Y)}(i) = \varsigma^2_{h_\theta,(X,Y)}(i) \triangleq \mathbb{E}_\theta \left( h_\theta(i \mid X) - h(i \mid X) \right)^2 , \tag{5}$$

$$\text{BVG}_{h_\theta,(X,Y)}(i) \triangleq \text{Bias}^2_{h_\theta,(X,Y)}(i) - \text{Vari}_{h_\theta,(X,Y)}(i) . \tag{6}$$

Throughout this paper, we use $\text{Bias}_{h_\theta,(X,Y)}(i)$ and $\beta_{h_\theta,(X,Y)}(i)$ interchangeably as synonyms, and we call $\varsigma_{h_\theta,(X,Y)}(i)$ the standard deviation of the $i$-th entry. Moreover, the total bias, variance, and bias-variance gap are defined as

$$\text{Bias}_{h_\theta,(X,Y)} \triangleq \sqrt{\sum_{i \in [K]} \text{Bias}^2_{h_\theta,(X,Y)}(i)} = \|h(\cdot \mid X) - e_Y\|_2 , \tag{7}$$

$$\text{Vari}_{h_\theta,(X,Y)} \triangleq \sum_{i \in [K]} \text{Vari}_{h_\theta,(X,Y)}(i) = \mathbb{E}_\theta \|h_\theta(\cdot \mid X) - h(\cdot \mid X)\|^2_2 , \tag{8}$$

$$\text{BVG}_{h_\theta,(X,Y)} \triangleq \text{Bias}^2_{h_\theta,(X,Y)} - \text{Vari}_{h_\theta,(X,Y)} . \tag{9}$$

where $e_i \in \mathbb{R}^K$ is a vector whose $i$-th entry is 1 and all other entries are 0.

It is well-known that bias and variance provide a decomposition of the expected risk with respect to the MSE loss (Hastie et al., 2009, equation (2.25)). That is,

$$\text{Risk}_{h_\theta,(X,Y)} \triangleq \mathbb{E}_\theta \|h_\theta(\cdot \mid X) - e_Y\|^2_2 = \text{Bias}^2_{h_\theta,(X,Y)} + \text{Vari}_{h_\theta,(X,Y)} . \tag{10}$$

---

[1]In all of our empirical results, the randomness comes from weight initialization and bootstrapping of training set, following Neal et al. (2018). The effect of different sources of randomness is studied in Appendix E.6.

| Premise | Finding | Pointwise vs. In aggregate? | Empirical vs. Theoretical? | References |
|---|---|---|---|---|
| Calibration | Generalization error = Disagreement | In aggregate | Both | (Jiang et al., 2022; Kirsch & Gal, 2022) |
| Generalization | Calibration | In aggregate | Empirical | (Carrell et al., 2022) |
| Multi-domain calibration | Out-of-domain generalization | In aggregate | Empirical | (Wald et al., 2021) |
| **Calibration** | $\text{Bias}^2 \approx \text{Variance}$ | **Pointwise** | **Both** | **This work** |
| **Neural collapse** | $\text{Bias}^2 \approx \text{Variance}$ | **Pointwise** | **Both** | **This work** |

Table 2: Summary of findings about calibration, generalization and disagreements.

We focus on models trained using the cross-entropy (CE) loss throughout the paper. However, we analyze the bias and variance of these models using a decomposition of the mean squared error (MSE) loss. We present results for bias and variance from decomposing CE loss in Section E.2.

We now introduce several notations that will be used throughout the paper.

**Definition 2.2.** The *prediction*, *confidence*, and *accuracy* of $h$ on $x$ are defined by

$$\text{pred}_h(x) = \arg\max_{j \in [K]} h(j \mid x), \ \ \text{conf}_h(x) = h(\text{pred}_h(x) \mid x), \ \ \text{acc}_h(x) = \mathbb{P}_{Y|X}(\text{pred}_h(x) \mid x).$$

The *uncertainty* of the ensemble $\{h_\theta\}$ on $x$ is $\text{Unce}_{h_\theta}(x) = 1 - \mathbb{E}_\theta \|h_\theta(\cdot \mid x)\|_2^2$.

## 2.2 RELATED WORK

**Bias-variance decomposition in deep learning.** In the classical statistical learning theory of bias-variance tradeoff, increasing the model capacity beyond a certain point leads to overfitting (Geman et al., 1992). However, deep neural networks in practice usually contain a large number of parameters but still generalize well. Towards bridging the gap between theory and practice, one of the most famous work is Belkin et al. (2019) which reveals a "double-descent" curve to subsume the U-shaped tradeoff curve. This surprising observation motivates the work of Neal et al. (2018) to measure the bias and variance in popular deep models, leading to a discovery of a "unimodal variance" phenomenon (Yang et al., 2020). Subsequent work includes Adlam & Pennington (2020); Lin & Dobriban (2021) that study variance under fine-grained decompositions, and Rocks & Mehta (2022a;b) that analyze variance under simplified regression or random feature models.

**Calibration.** Calibration is a fundamental quantity in machine learning which informally speaking measures the degree to which the output distribution from a model agrees with the Bayes probability of the labels over the data (Guo et al., 2017). Previous works proposed a theory on calibration implying the generalization-disagreement equality (Jiang et al., 2022; Kirsch & Gal, 2022). In Table 2 we summarize several related results connecting calibration with other fundamental concepts from previous works. Our work can be viewed as extending these works as we connect calibration and the bias-variance alignment.

**Neural collapse.** Towards understanding last layer features learned in deep network based classification models, the work of Papyan et al. (2020) reveals the *neural collapse* phenomenon that offers a clear mathematical characterization: Within-class features collapse to their corresponding class means, and between-class separation of the class means is maximized. This observation motivates a sequence of theoretical work on justifying its occurrence (Fang et al., 2021; Zhu et al., 2021; Tirer & Bruna, 2022; Poggio & Liao, 2020; Thrampoulidis et al., 2022), and practical work on leveraging the insights to improve model performance (Liang & Davis, 2023; Yang et al., 2023; Li et al., 2022).

## 3 EMPIRICAL ANALYSIS OF BIAS-VARIANCE ALIGNMENT

We begin by providing a quantitative measure in Section 3.1 on the alignment of bias and variance illustrated in Figure 1b. Then, we provide empirical evidence in Section 3.2 that the bias-variance alignment phenomenon occurs more prevalently for networks beyond ResNets, and for datasets other than ImageNet. Finally, in Section 3.3 we study the effect of network size, showing that bias-variance alignment is a phenomenon for over-parameterized models.

## 3.1 QUANTITATIVE REGRESSION ANALYSIS OF BIAS-VARIANCE ALIGNMENT

| Model name | $R^2$ | Slope |
|---|---|---|
| ResNet-8 (CIFAR-10) | 0.979 | 0.882 |
| ResNet-56 (CIFAR-10) | 0.996 | 0.964 |
| ResNet-110 (CIFAR-100) | 0.986 | 0.901 |
| ResNet-50 (ImageNet) | 0.977 | 0.897 |

Table 3: Coefficient of determination ($R^2$) and slope for linear regression of $\log \mathrm{Vari}_{h_\theta,(x_i,y_i)}$ on $\log \mathrm{Bias}_{h_\theta,(x_i,y_i)}$.

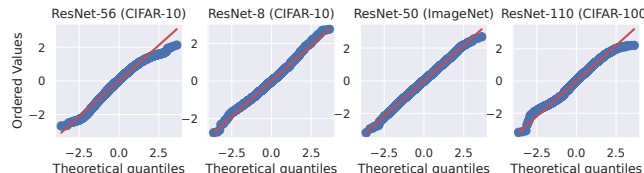

Figure 2: The Q-Q plots of the residuals of linear regression of $\log \mathrm{Vari}_{h_\theta,(x_i,y_i)}$ on $\log \mathrm{Bias}_{h_\theta,(x_i,y_i)}$ of the four models.

We start by conducting a quantitative regression analysis of bias and variance on the logarithmic scale for verifying the bias-variance alignment phenomenon in Eq. (1). First, in Table 3, we present the coefficient of determination $R^2$ and the slope of the linear regression of $\log \mathrm{Vari}_{h_\theta,(x_i,y_i)}$ on $\log \mathrm{Bias}_{h_\theta,(x_i,y_i)}$ for the following models and datasets: ResNet-56 (on CIFAR-10), ResNet-8 (on CIFAR-10), ResNet-50 (on ImageNet), and ResNet-110 (on CIFAR-100). Notice how the coefficient of determination for all four settings is extremely close to 1 (at least 0.977) and the slope is also very close to 1, demonstrating a very strong linear alignment of the two quantities. We next analyze the normality of the residuals of the logarithmic linear regression. The Q-Q plots are shown in Figure 2. We observe that the residuals of the linear regression on the four models are all approximately normal, especially for the data points whose sample quantile is between $-1.5$ and $1.5$.

## 3.2 PREVALENCE OF BIAS-VARIANCE ALIGNMENT ACROSS ARCHITECTURES AND DATASETS

In Figure 1b, we showed that the bias-variance alignment occurs for ResNet-50 trained on ImageNet. Here, we provide additional evidence in Figure 3 that the bias-variance alignment occurs for other model architectures and datasets.

In particular, Figure 3a and Figure 3b show the sample-wise bias-variance of EfficientNet-B0 (Tan & Le, 2019) and MobileNet-V2 (Sandler et al., 2018), respectively, trained on the ImageNet dataset. Figure 3c and Figure 3d show the sample-wise bias-variance of ResNet-110 trained on CIFAR-10 and CIFAR-100, respectively. In all cases, we observe a similar pattern, demonstrating the prevalence of the bias-variance alignment with respect to network architectures and choice of dataset. Finally, in Figure 12 we confirm the observation on the NLP benchmark, where we finetune BERT models on TREC the dataset.

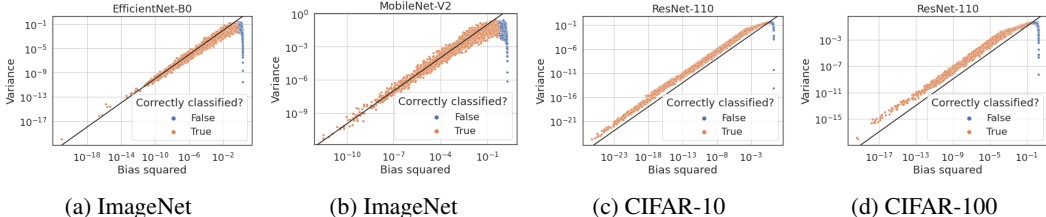

| (a) ImageNet | (b) ImageNet | (c) CIFAR-10 | (d) CIFAR-100 |
|---|---|---|---|

Figure 3: Sample-wise bias and variance for *(a, b)*: Varying model architectures trained on the ImageNet dataset, and *(c, d)*: Two additional datasets, namely CIFAR-10 and CIFAR-100. For ImageNet, CIFAR-10 and CIFAR-100, bias and variance are estimated from 10, 100 and 100 independently trained models, respectively.

## 3.3 ROLE OF OVER-PARAMETERIZATION

We investigate the impact of model size on the bias-variance alignment phenomenon, demonstrating that it only occurs for large and over-parameterized models.

To vary the size of the model, we consider ResNets with depths of 8, 20, 56, and 110, which we denote as ResNet-$\{8, 20, 56, 110\}$. To obtain even smaller models, we reduce the width of ResNet-8 from 16 to 8, 4, 2, and 1. Here, the width refers to the number of filters in the convolutional layers of

ResNet-8. Specifically, the convolutional layers of ResNet-8 can be divided into three stages with 16, 32, and 64 filters, respectively. Therefore, ResNet-8 with width $k$, denoted as ResNet-8-W[$k$], refers to a ResNet-8 with $k$, $2k$, and $4k$ filters in the three stages, respectively.

Due to space limit, we choose three models, namely ResNet-8-W1, ResNet-8-W16, and ResNet-110-W16 from the eight model sizes discussed above, and present their sample-wise bias and variance evaluated on CIFAR-10 in Figures 4a to 4c. These three models represent a small, medium, and large size model, respectively. Results for all the other model sizes are provided in Figure 5 (see Appendix). We can see that bias-variance alignment becomes increasingly pronounced with larger models.

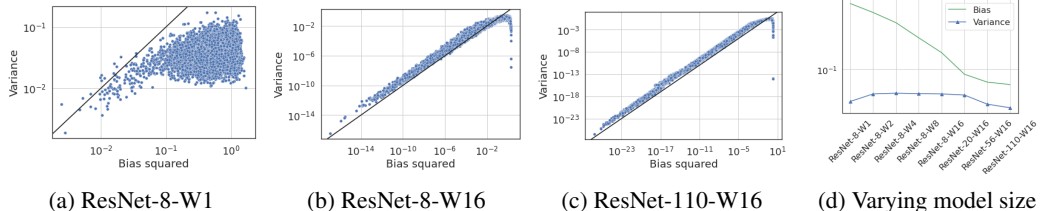

(a) ResNet-8-W1      (b) ResNet-8-W16      (c) ResNet-110-W16      (d) Varying model size

Figure 4: Figures 4a to 4c: Sample-wise bias and variance for networks of varying scale trained on CIFAR-10. Here, ResNet-[$p$]-W[$q$] refers to ResNet with $p$ layers and width $q$. The model size monotonically increases from the leftmost figure to the rightmost figure. For all cases, the bias and variance are estimated from 50 independently trained models. Figure 4d: Averaged bias and variance over all test samples under varying model sizes.

The results in Figure 4 suggest that the emergence of bias-variance alignment is associated with *over-parameterization*, which refers to large capacity deep models that can perfectly fit any training data. Classical theory suggests that bias and variance exhibit a trade-off relation, where larger model size reduces bias and increases variance. However, Yang et al. (2020) shows that this trade-off holds only in the regime where the model size is relatively small. For over-parameterized models, the variance does not continue to grow; rather, it exhibits a *unimodal* shape. To examine bias-variance alignment in association with the unimodal variance phenomenon of Yang et al. (2020), we compute the averaged bias and variance over all test samples for results reported in Figures 4a to 4c (and also results of five additional models reported in Figure 5), and plot them as a function of model size in Figure 4d. The result in Figure 4d aligns with the observation in Yang et al. (2020), namely, bias is monotonically decreasing and variance curves is unimodal. Moreover, the model that exhibits strong bias-variance alignment in Figure 4, namely ResNet-56-W16, clearly is outside of the classical regime of bias-variance tradeoff shown in Figure 4d. Meanwhile, model in the classical trade-off regime, e.g. ResNet-8-W1, does not have bias-variance alignment.

## 4   CALIBRATION IMPLIES BIAS-VARIANCE ALIGNMENT

In this section we show how model calibration can imply bias-variance alignment. We start from unifying the different calibration views from previous work, and then show how each of these assumptions implies a different version of the bias-variance correlation.

### 4.1   A UNIFIED VIEW OF CALIBRATION

In previous work, various definitions of calibration have been introduced. In the following, we present a general definition that encompasses a wide range of these definitions. Suppose that there is a collection of sub-$\sigma$-algebras $\{\Sigma_i\}_{i \in [K]}$ of $\sigma(X)$, where $\sigma(X)$ is the $\sigma$-algebra generated by the random variable $X$. [2] In this section, we refer to the $\sigma$-algebra generated by a random variable or event using the notation $\sigma(\cdot)$.

---

[2]In Definition 4.1, the $\sigma$-algebras $\Sigma_i$ must be sub-$\sigma$-algebras of $\sigma(X)$. This is because we want the random variables and events we are conditioning on to be functions of $X$. Thus, the conditional expectation in the definition of calibration averages over all data examples that have the same value of a function of $X$.

| | $\text{ECE}_{\mathbb{P}}^{\{\Sigma_i\}_{i \in [K]}}$ | $\text{CWCE}_{\mathbb{P}}^{\{\Sigma_i\}_{i \in [K]}}$ |
|---|---|---|
| $\Sigma_i^{\text{samp}}$ | (Kirsch & Gal, 2022, Expectation of Eq. (6)) | **This work (Definition 4.2)** |
| $\Sigma_i^{\text{pre}}$ | (Guo et al., 2017, Eq. (2)) **(see our Appendix F.2)** | (Kirsch & Gal, 2022, Eq. (36)) |
| $\Sigma_i^{\text{bin}}$ | (Guo et al., 2017, Eq. (3)) (Naeini et al., 2015) (Nixon et al., 2019) **(see our Appendix F.3)** | **This work (Definition 4.2)** |

Table 4: Summary of the various definitions of $\text{ECE}_{\mathbb{P}}^{\{\Sigma_i\}_{i \in [K]}}$ and $\text{CWCE}_{\mathbb{P}}^{\{\Sigma_i\}_{i \in [K]}}$ that have been proposed in previous work. Our unified definition subsumes these previous definitions.

**Definition 4.1** (Perfect (confidence) calibration). A function $h : \mathcal{X} \to \mathcal{M}([K])$ has perfect calibration with respect to $\{\Sigma_i\}_{i \in [K]}$ if the following equation holds for all $i \in [K]$:

$$\mathbb{E}[\Delta(i \mid X) \mid \Sigma_i] = 0, \quad \Delta(i \mid x) \triangleq h(i \mid x) - \mathbb{P}_{Y|X}(i \mid x). \tag{11}$$

The function has perfect confidence calibration if the following equation holds:

$$\mathbb{E}[\Delta(\text{pred}_h(X) \mid X) \mid \Sigma_{\text{pred}_h(X)}] = \mathbb{E}[(\text{conf}_h(X) - \text{acc}_h(X)) \mid \Sigma_{\text{pred}_h(X)}] = 0. \tag{12}$$

The sub-$\sigma$-algebras $\{\Sigma_i\}_{i \in [K]}$ control the granularity of perfect calibration. In other words, (11) in Definition 4.1 says that $\Delta(i \mid X)$ vanishes on average, and $\{\Sigma_i\}_{i \in [K]}$ specifies the set of samples that we average over. Here are examples of ways of choosing $\{\Sigma_i\}_{i \in [K]}$:

- Sample-wise perfect calibration $\Sigma_i^{\text{samp}} = \sigma(X)$. In this case, Definition 4.1 is equivalent to $h(i \mid X) = \mathbb{P}_{Y|X}(i \mid X)$ for every $i \in [K]$. In other words, perfect calibration holds for every sample $X$.
- Pre-image perfect calibration $\Sigma_i^{\text{pre}} = \sigma(h(i \mid X))$. This may be the most widely used definition of calibration (Guo et al., 2017; Kirsch & Gal, 2022) and characterizes the calibration averaged over all samples that share a common prediction $h(\cdot \mid X)$.
- Bin-wise perfect confidence calibration $\Sigma_i^{\text{bin}} = \sigma(\mathbf{1}\{\lceil Mh(i \mid X)\rceil\})$ where $M$ is a positive integer (Guo et al., 2017). The map $x \mapsto \lceil Mh(i \mid x)\rceil$ assigns $x$ to $M$ bins (if $h(i \mid x) > 0$ for all $x$): $(0, \frac{1}{M}], (\frac{1}{M}, \frac{2}{M}], \ldots, (\frac{M-1}{M}, 1]$ according to the value of $h(i \mid x)$. Then the samples that fall into the same bin are averaged over.

**Definition 4.2** (Calibration Errors). If we define $\Delta(i \mid x)$ as in (11), the expected calibration error ($\text{ECE}_{\mathbb{P}}^{\{\Sigma_i\}_{i \in [K]}}$) and the class-wise calibration error ($\text{CWCE}_{\mathbb{P}}^{\{\Sigma_i\}_{i \in [K]}}$) of a function $h : \mathcal{X} \to \mathcal{M}([K])$ with respect to $\{\Sigma_i\}_{i \in [K]}$ on the data distribution $\mathbb{P} \in \mathcal{M}(\mathcal{X} \times \mathcal{Y})$ are given by

$$\text{ECE}_{\mathbb{P}}^{\{\Sigma_i\}_{i \in [K]}}(h) \triangleq \mathbb{E}\left|\mathbb{E}[\Delta(\text{pred}_h(X) \mid X) \mid \Sigma_{\text{pred}_h(X)}]\right|,$$

$$\text{CWCE}_{\mathbb{P}}^{\{\Sigma_i\}_{i \in [K]}}(h) \triangleq \sum_{i \in [K]} \mathbb{E}\left|\mathbb{E}[\Delta(i \mid X) \mid \Sigma_i]\right|.$$

We elucidate how our unified definition subsumes the various definitions in previous work, and we summarize it in Table 4. In Appendix F.2 and Appendix F.3, we show concretely what $\text{ECE}_{\mathbb{P}}^{\{\Sigma_i\}_{i \in [K]}}$ looks like with respect to $\Sigma_i^{\text{pre}}$ and $\Sigma_i^{\text{bin}}$, respectively.

## 4.2 CALIBRATION MEETS THE BIAS-VARIANCE DECOMPOSITION

In this subsection we then show that, under the model calibration assumption, there is a correlation between squared bias and variance. Moreover, in the case of imperfect model calibration, the discrepancy between the squared bias and variance can be bounded by the calibration error.

Recall the definitions of bias and variance in Section 2.1. Theorem 4.3 characterizes the discrepancy between the squared bias and variance, and provides an upper bound for it by the class calibration error. Corollary 4.4, which follows the theorem, shows that in the case of perfect calibration, the variance is upper bounded by the squared bias. Moreover, if the model outputs a completely certain prediction (outputs a one-hot vector), it is theoretically guaranteed that the squared bias equals variance, i.e., the bias-variance correlation appears. If the model does not output a completely certain prediction but a highly certain prediction, an approximate bias-variance correlation follows.

**Theorem 4.3.** *If $\{h_\theta\}$ is an ensemble whose mean function $h(i \mid X)$ is $\Sigma_i$-measurable, then we have*

$$\mathbb{E}_{Y|X}\left[\text{BVG}_{h_\theta, (X,Y)}(i)\right] = 2h(i \mid X)\Delta(i \mid X) + \mathbb{P}_{Y|X}(i \mid X) - \mathbb{E}_\theta h_\theta(i \mid X)^2,$$

$$\mathbb{E}\left|\mathbb{E}\left[\mathbb{E}_{Y|X}[\text{BVG}_{h_\theta, (X,Y)}(i)] - \mathbb{P}_{Y|X}(i \mid X) + \mathbb{E}_\theta h_\theta(i \mid X)^2 \mid \Sigma_i\right]\right| \leq 2\,\text{CCE}_{\mathbb{P}}^{\{\Sigma_i\}_{i \in [K]}}(i),$$

| Width factor | LHS of (13) | RHS of (13) | $\mathrm{CWCE}_{\mathbb{P}}^{\{\Sigma_i^{\mathrm{bin}}\}}$ |
|:---:|:---:|:---:|:---:|
| $1/2$ | 0.64 | 0.90 | 0.45 |
| $1/4$ | 0.63 | 0.74 | 0.37 |
| $1/8$ | 0.66 | 0.70 | 0.35 |
| $1/16$ | 0.76 | 0.90 | 0.45 |

Table 5: Empirical summary of values for $\mathrm{CWCE}_{\mathbb{P}}^{\{\Sigma_i^{\mathrm{bin}}\}}$ and the bias and variance, both calculated w.r.t. $\Sigma_i^{\mathrm{bin}}$ (where we use 20 equally spaced bins) from ResNet-8 models on CIFAR-10 across varying width.

where $\mathrm{CCE}_{\mathbb{P}}^{\{\Sigma_i\}_{i\in[K]}}(i) \triangleq \mathbb{E}\left|\mathbb{E}[\Delta(i\mid X)\mid \Sigma_i]\right|$ *is the class calibration error for class* $i$. *If* $\Sigma = \bigcap_{i\in[K]}\sigma(\Sigma_i)$, *for total squared bias and variance, the following equations holds*

$$\mathbb{E}\left[\mathbb{E}_{Y|X}[\mathrm{BVG}_{h_\theta,(X,Y)}]\mid \Sigma\right] = \mathbb{E}\left[\mathrm{Unce}_{h_\theta}(X)\mid \Sigma\right] + 2\sum_{i\in[K]}\mathbb{E}\left[\mathbb{E}[\Delta(i\mid X)\mid \Sigma_i]h(i\mid X)\mid \Sigma\right],$$

$$\left|\mathbb{E}\left[\mathbb{E}_{Y|X}[\mathrm{BVG}_{h_\theta,(X,Y)}] - \mathrm{Unce}_{h_\theta}(X)\mid \Sigma\right]\right| \leq 2\,\mathrm{CWCE}_{\mathbb{P}}^{\{\Sigma_i\}_{i\in[K]}} \tag{13}$$

We observe that, without any additional assumptions, $h(i|X)$ is automatically $\Sigma_i^{\mathrm{samp}}$- and $\Sigma_i^{\mathrm{pre}}$-measurable. If one chooses $\Sigma_i^{\mathrm{samp}} = \sigma(X)$, then $\mathbb{E}[\mathbb{E}_{Y|X}[\cdot]\mid \Sigma] = \mathbb{E}_{Y|X}[\cdot]$. In this case, Theorem 4.3 holds for every example $x\in\mathcal{X}$.

**Corollary 4.4.** *If* $h$ *has perfect calibration with respect to* $\{\Sigma_i\}_{i\in[K]}$, *then* $\mathbb{E}[\Delta(i\mid X)\mid \Sigma_i] = 0$ *and therefore we have*

$$\mathbb{E}\left[\mathbb{E}_{Y|X}[\mathrm{BVG}_{h_\theta,(X,Y)}(i)]\mid \Sigma_i\right] = \mathbb{E}\left[\mathbb{P}_{Y|X}(i\mid X) - \mathbb{E}_\theta h_\theta(i\mid X)^2\mid \Sigma_i\right]$$

$$= \mathbb{E}\left[h(i\mid X) - \mathbb{E}_\theta h_\theta(i\mid X)^2\mid \Sigma_i\right] = \mathbb{E}\left[\mathbb{E}_\theta\left[h_\theta(i\mid X)(1 - h_\theta(i\mid X))\right]\mid \Sigma_i\right], \tag{14}$$

$$\mathbb{E}\left[\mathbb{E}_{Y|X}[\mathrm{BVG}_{h_\theta,(X,Y)}]\mid \Sigma\right] = \mathbb{E}\left[\mathrm{Unce}_{h_\theta}(X)\mid \Sigma\right] \geq 0. \tag{15}$$

*Moreover, if* $h_\theta$ *outputs a one-hot vector, we have* $\|h_\theta(\cdot\mid X)\|_2^2 = 1$ *and therefore* $\mathbb{E}\left[\mathbb{E}_{Y|X}[\mathrm{BVG}_{h_\theta,(X,Y)}]\mid \Sigma\right] = 0$. *In other words, in expectation* $\mathbb{E}[\mathbb{E}_{Y|X}[\cdot]\mid \Sigma]$, *the squared bias equals variance.*

Corollary 4.4 shows that if $h$ has perfect calibration, then in expectation $\mathbb{E}[\mathbb{E}_{Y|X}[\cdot]\mid \Sigma]$, the variance is upper bounded by the bias and the gap between them is $\mathrm{Unce}_{h_\theta}(X)$. If $h_\theta(\cdot\mid X)$ is highly confident (i.e., $\max_{i\in[K]}h_\theta(\cdot\mid X)\approx 1$), then $\mathrm{Bias}_{h_\theta,(X,Y)}^2\approx \mathrm{Vari}_{h_\theta,(X,Y)}$. The extreme case is that $h_\theta$ outputs a one-hot vector.

**Corollary 4.5.** *If* $h$ *has perfect calibration with respect to* $\{\Sigma_i\}_{i\in[K]}$ *and* $h_\theta(i\mid X)\to a$ *for every* $\theta$ *(a is either* 0 *or* 1*), then* $\mathbb{E}\left[\mathbb{E}_{Y|X}\left[\beta_{h_\theta,(X,Y)}(i) - \varsigma_{h_\theta,(X,Y)}(i)\right]\mid \Sigma_i\right]\to 0$.

Corollary 4.5 shows that when $h_\theta(i\mid X)$ is highly confident ($h_\theta(i\mid X)\to a\in\{0,1\}$), then $\beta_{h_\theta,(X,Y)}(i) - \varsigma_{h_\theta,(X,Y)}(i)\to 0$ in the mean $\mathbb{E}\left[\mathbb{E}_{Y|X}[\cdot]\mid \Sigma_i\right]$, which is entrywise bias-variance correlation. Moreover, since

$$\mathbb{E}\left[\mathbb{E}_{Y|X}\left[\beta_{h_\theta,(X,Y)}(i) - \varsigma_{h_\theta,(X,Y)}(i)\right]\mid \Sigma_i\right]$$
$$= \mathbb{E}[\mathbb{E}_{Y|X}[\mathbf{1}\{Y = i\}(1 - h(i\mid X) - \varsigma_{h_\theta,(X,Y)}(i)) + \mathbf{1}\{Y \neq i\}(h(i\mid X) - \varsigma_{h_\theta,(X,Y)}(i))]\mid \Sigma_i],$$

if $\mathbb{P}_{Y|X}(Y = i\mid \Sigma_i)\approx 1$, we have $h(i\mid X)\approx 1 - \varsigma_{h_\theta,(X,Y)}(i)$; and if $\mathbb{P}_{Y|X}(Y = i\mid \Sigma_i)\approx 0$, we have $h(i\mid X)\approx \varsigma_{h_\theta,(X,Y)}(i)$.

**Experiments confirm our theory.** We now present empirical results that support our theory. We empirically verify the inequality (13) across models of varying widths when using $\Sigma_i^{\mathrm{bin}}$ (where we use 20 equally spaced bins) for the definitions of calibration, uncertainty, bias and variance. Note that calibration requires estimating the true probability distribution, and so direct computation for $\Sigma_i^{\mathrm{pred}}$ or $\Sigma_i^{\mathrm{samp}}$ is infeasible. In Table 5 we empirically verify the relationship between bias, variance, and uncertainty when using $\Sigma_i^{\mathrm{bin}}$. The left-hand side of (13), which is $\left|\mathbb{E}\left[\mathbb{E}_{Y|X}[\mathrm{BVG}_{h_\theta,(X,Y)}] - \mathrm{Unce}_{h_\theta}(X)\mid \Sigma\right]\right|$, is computed from the bias and variance values. The right-hand side of (13) is $2\,\mathrm{CWCE}_{\mathbb{P}}^{\{\Sigma_i^{\mathrm{bin}}\}_{i\in[K]}}$. We see that across architectures, Equation (13) of Theorem 4.3 holds.

## 5 NEURAL COLLAPSE IMPLIES APPROXIMATE BIAS-VARIANCE ALIGNMENT

Neural collapse (Papyan et al., 2020) is a phenomenon pertaining the last layer features and classifier weights of a trained deep classification model. This section considers a statistical modeling of the

prediction of the network ensemble $\{h_\theta\}$ on an arbitrary test data $(X, Y)$ that is motivated from neural collapse, upon which we derive a bound on the ratio between $\text{Bias}^2_{h_\theta,(X,Y)}$ and $\text{Vari}_{h_\theta,(X,Y)}$.

**Modeling assumption motivated by neural collapse.** We assume that each model $h_\theta$ in the ensemble can be written as $h_\theta(\cdot \mid x) = \text{softmax}(W\psi_\tau(x))$, where $\theta = (\tau, W)$ denotes trainable parameters. In the above, we refer to $\psi_\tau(x) \in \mathbb{R}^d$ as the *feature* vector, and $W$ as the classifier weight. The *neural collapse* phenomenon states that during training, $W$ converges to a rotated and scaled version of the simplex equiangular tight frame (ETF) matrix $W^{\text{ETF}}$. That is,

$$W \propto W^{\text{ETF}} R^\top, \text{ where } W^{\text{ETF}} = \left(w_1^{\text{ETF}}, \ldots, w_K^{\text{ETF}}\right) \triangleq \sqrt{\frac{K}{K-1}} \left(I_K - \frac{1}{K}\mathbf{1}_K\mathbf{1}_K^\top\right), \quad (16)$$

and $R \in \mathbb{R}^{d \times K}$ is an orthogonal matrix (i.e., $R^\top R = I_K$). In above, $I_K \in \mathbb{R}^{K \times K}$ is an identity matrix, and $\mathbf{1}_K \in \mathbb{R}^K$ is a column vector with all entries being 1. We summarize the properties of $W^{\text{ETF}}$ in Appendix G.1. Moreover, for any *training* data $(x^{\text{train}}, y^{\text{train}})$, neural collapse predicts that the feature $\psi_\tau(x^{\text{train}})$ is aligned with its classifier weight, i.e., $\psi_\tau(x^{\text{train}}) = sRw_{y^{\text{train}}}^{\text{ETF}}$ for some $s > 0$ independent of $(x^{\text{train}}, y^{\text{train}})$.

For a test sample $(X, Y)$, neural collapse does not predict the distribution of its feature $\psi_\tau(X)$. However, it is reasonable to assume that it slightly deviates from the training feature of class $Y$. This motivates us to assume that $\psi_\tau(X) = R\left(sw_Y^{\text{ETF}} + v\right)$, where $v$ is the noise vector, which leads to $h_\theta(\cdot \mid X) = \text{softmax}(W^{\text{ETF}}(sw_Y^{\text{ETF}} + v))$. Hence, the prediction of the network ensemble $\{h_\theta\}$ may be modeled as follows.

**Assumption 5.1.** We assume $\{h_\theta(\cdot \mid X)\} = \{\text{softmax}\left(W^{\text{ETF}}(sw_Y^{\text{ETF}} + v)\right)\}$ for any test sample $(X, Y)$, where $v$ has i.i.d. entries drawn according to $-\beta\sqrt{\frac{K}{K-1}}v_i \sim \text{Gumbel}(\mu, \beta)$.

Appendix G.2 shows that the assumption on $v$ aligns with the observation in practical networks.

**Bias-variance analysis.** Theorem 5.2 computes the entrywise bias and standard deviation under Assumption 5.1, for entries corresponding to the true class.

**Theorem 5.2.** *Consider the model ensemble in Assumption 5.1. Let $K' = K - 1$, $c = \frac{e^{sK/K'}}{K'}$ and*

$$\phi_{K'}(c) \triangleq \frac{c^{K'-1}\left(c - \frac{1}{K'}\right)^{-K'}(cK' - K'\log(cK') - 1)}{cK' - 1} + \frac{\left(c - \frac{1}{K'}\right)^{-K'-1}}{K'}\sum_{j=1}^{K'-1}\frac{(K'-j)(c-1/K')^j c^{-j+K'-1}}{j}.$$

*Then we have $\beta_{h_\theta,(X,Y)}(Y) = |c\phi_{K'}(c) - 1|$ and $\varsigma_{h_\theta,(X,Y)}(Y) = c\sqrt{-\left(\frac{d\phi_{K'}(c)}{dc} + \phi_{K'}(c)^2\right)}$, where $\beta_{h_\theta,(X,Y)}(\cdot)$ and $\varsigma_{h_\theta,(X,Y)}(\cdot)$ are defined in (4) of Definition 2.1.*

Due to technical difficulties, Theorem 5.2 does not provide bias and variance for entries other than those corresponding to the true class. However, under the special case of binary classification, we are able to provide bias and variance for all entries as follows.

**Corollary 5.3.** *If $K = 2$, then for $i \in \{1, 2\}$ we have $\beta_{h_\theta,(X,Y)}(i) = \frac{|\log(c)c - c + 1|}{(c-1)^2}$, $\varsigma_{h_\theta,(X,Y)}(i) = \frac{\sqrt{c((c-1)^2 - c\log^2(c))}}{(c-1)^2}$, where $c = e^{2s}$. Furthermore, we have: (1) on the linear scale, $\frac{\beta_{h_\theta,(X,Y)}(i)}{\varsigma_{h_\theta,(X,Y)}(i)}$ is a decreasing function of $s$ and $1.74 > \frac{\beta_{h_\theta,(X,Y)}(i)}{\varsigma_{h_\theta,(X,Y)}(i)} \geq \frac{2s-1}{e^s}$, or equivalently, $1.74^2 > \frac{\text{Bias}^2_{h_\theta,(X,Y)}}{\text{Vari}_{h_\theta,(X,Y)}} \geq \frac{(2s-1)^2}{e^{2s}}$; (2) on the logarithmic scale, $\frac{\log\text{Bias}_{h_\theta,(X,Y)}(i)}{\log\text{Vari}_{h_\theta,(X,Y)}(i)} \in (0.557, 2)$.*

Corollary 5.3 illustrates the entrywise bias and standard deviation for binary classification. We prove the approximate correlation between the entrywise bias and standard deviation by providing an upper and lower bound for the ratio of the entrywise bias to the standard deviation.

## 6 CONCLUSION

We show that bias and variance align at a sample level for ensembles of deep learning models, suggesting a more nuanced bias-variance relation in deep learning. We study this phenomenon from two theoretical perspectives, calibration and neural collapse, and provide new insights into the bias-variance alignment.

## 7 ACKNOWLEDGEMENTS

We would like to acknowledge helpful comments from Aditya Krishna Menon (Google Research) and Christina Baek (Carnegie Mellon University).

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

# Appendix

## Table of Contents

## A  SOCIETAL IMPACT STATEMENT

This paper aims to expose a peculiar bias-variance alignment phenomenon and characterize it both theoretically and empirically. We do not foresee any negative societal consequences from this work.

## B  LIMITATIONS

We identify the following limitations of our work:

- We only considered the squared loss and cross entropy loss functions. It would be interesting to extend our results to other loss functions, such as the 0/1 loss.

- Our theory is based on the binary classification assumption. We plan to extend it to multi-class classification in future work.
- It would be interesting to study the bias-variance alignment theoretically in an end-to-end manner, using tools such as the neural tangent kernel theory or a mean-field analysis.
- We conducted our experiments in the image classification domain. It would be interesting to verify our findings in other domains, such as natural language processing (NLP).

We believe that these limitations do not detract from the overall significance of our work. Our findings provide new insights into the bias-variance alignment phenomenon, and we hope that this paper will stimulate further research on this important topic.

## C  FURTHER RELATED WORK.

It was shown that statistics such as accuracy (Miller et al., 2021) and disagreement (Baek et al., 2023) are highly correlated when contrasted across in-distribution and out-distribution data. This points at a potential extension to our work to consider how our findings translate to the out of domain data.

Gupta et al. (2022) investigate the theoretical underpinnings of ensemble methods for classification tasks, extending the bias-variance decomposition to derive generalized laws of total expectation and variance for nonsymmetric losses. Their work sheds light on the mechanisms by which ensembles reduce variance and potentially bias, providing valuable insights for improving the performance of ensemble classifiers. Ortega et al. (2022) provide a comprehensive theoretical framework for understanding the relationship between diversity and generalization error in neural network ensembles. They analyze the impact of diversity on ensemble performance for various loss functions and model combination strategies, offering valuable insights for designing effective ensemble learning algorithms. Brown et al. (2005) focus on managing diversity in regression ensembles. They introduce a control mechanism through the error function, demonstrating its effectiveness in improving ensemble performance over traditional methods. This work provides insights into systematic control of the bias-variance-covariance trade-off in regression ensembles. Abe et al. (2023) challenge conventional wisdom on predictive diversity in deep neural network ensembles. While diversity benefits small models, the authors find that encouraging diversity harms high-capacity deep ensembles used for classification. Their experiments show that diversity-encouraging regularizers hinder performance, suggesting that the best strategy for deep ensembles may involve using more accurate but less diverse component models. In contrast to traditional ensemble methods, deep ensembles of neural networks offer the potential for direct optimization of overall performance. However, Jeffares et al. (2023) reveal that jointly minimizing ensemble loss induces base learners to collude, inflating artificial diversity. This pseudo-diversity fails to generalize, highlighting limitations in direct joint optimization and its impact on generalization gaps.

## D  PRACTICAL APPLICATIONS.

We believe that our findings on the bias-variance alignment can be used to develop new methods for validating deep learning models and selecting generalizable models in practice. One practical application is estimating the test error of a deep learning model using variance. This is possible because our finding is that bias and variance are aligned, and so we can estimate bias from variance. This means that even when the true labels of the test data are unavailable, we can still get a good estimate of the test error by measuring variance over multiple models on the test data. Compared to Jiang et al. (2022) which analyzed the alignment between disagreement and test error across the entire dataset, our method is a per-example approach and thus enables example-level validation of deep learning models. This is a novel way of validating deep learning models and selecting generalizable models in practice even when the true labels of the test data are unavailable. Additionally, our method is simple to implement, so it could be easily adopted by practitioners. Moreover, inspired by our result, one could consider practical algorithms leveraging the observation of bias and variance alignment. As one example, one can consider routing between ensembles of models. Given two ensembles of models, one could dynamically route between such two ensembles, depending which one yields lower variance. Given the above possible applications, we believe that our work has the potential to make a significant contribution to the field of deep learning.

# E MORE EMPIRICAL ANALYSIS OF BIAS-VARIANCE ALIGNMENT

## E.1 ADDITIONAL RESULTS ON ROLE OF OVER-PARAMETERIZATION

In Section 3.3 we showed that the bias-variance alignment phenomenon becomes more pronounced for over-parameterized models, by plotting sample-wise bias-variance for three models of varying sizes in Figure 4. Here we present results on five additional models of varying size Figure 5 that complement the results in Figure 4.

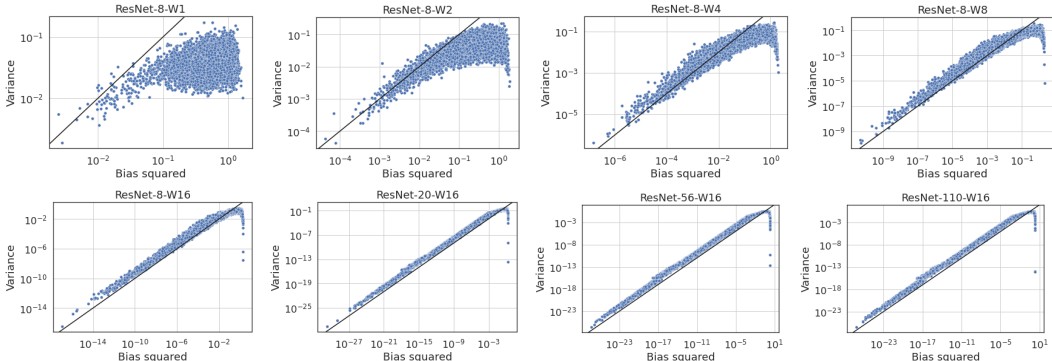

Figure 5: Sample-wise bias and variance for networks of varying scale trained on CIFAR-10.

## E.2 BIAS-VARIANCE DECOMPOSITION OF CROSS-ENTROPY LOSS

Deep neural networks for classification tasks are typically trained with the cross-entropy (CE) loss. Here, we investigate whether the bias and variance from decomposing the CE loss also exhibit the alignment phenomenon. The risk with respect to the CE loss can be decomposed as follows (Pfau, 2013):

$$\underbrace{\mathbb{E}_\theta\langle e_Y, \log(h_\theta(\cdot \mid X))\rangle}_{\text{Risk}} = \underbrace{D_{\text{KL}}(e_Y \parallel \bar{h}(\cdot \mid X))}_{\text{Bias}^2} + \underbrace{\mathbb{E}_\theta D_{\text{KL}}(\bar{h}(\cdot \mid X) \parallel h_\theta(\cdot \mid X)))}_{\text{Variance}}, \qquad (17)$$

where $D_{\text{KL}}$ denotes the KL divergence. In the above equation, $\bar{h}(\cdot|X)$ is obtained by taking the expectation of the log-probabilities and then applying a softmax function. In other words,

$$\bar{h}(i \mid X)) = \frac{\exp \mathbb{E}_\theta \log(h_\theta(i \mid X)))}{\sum_{i'} \exp \mathbb{E}_\theta \log(h_\theta(i' \mid X)))} \qquad (18)$$

Intuitively, $\bar{h}(\cdot|X)$ represents the average prediction under the KL divergence, assigning a probability proportional to $\exp \mathbb{E}_\theta \log(h_\theta(i \mid X))$ to each class $i$. The bias term measures the KL divergence between the true distribution $e_Y$ and the average prediction $\bar{h}(\cdot|X)$, quantifying the deviation of the ensemble's mean prediction from the actual class distribution. The variance term, on the other hand, captures the average KL divergence between the individual predictions in the ensemble and the mean prediction $\bar{h}(\cdot|X)$, reflecting the overall variability of the ensemble's predictions.

In Figure 7 we present the sample-wise bias and variance from decomposing the CE loss under the same setup as that in Figure 1b. In other words, the only difference between Figure 7 and Figure 1b is that the bias and variance are computed from decomposing the CE and MSE loss, respectively. It can be seen that the bias no longer aligns well with the variance. In Appendix F.6, we theoretically explain this phenomenon.

## E.3 PERMUTATION TEST FOR THE RESIDUAL AND THE LOG-BIAS

We perform linear regression of $\log \text{Vari}_{h_\theta,(x_i,y_i)}$ on $\log \text{Bias}_{h_\theta,(x_i,y_i)}$ for the following models and datasets: ResNet-56 (on CIFAR-10), ResNet-8 (on CIFAR-10), ResNet-50 (on ImageNet), and ResNet-110 (on CIFAR-100). We would like to test whether the residual is linearly correlated of the exogenous variable $\log \text{Bias}_{h_\theta,(x_i,y_i)}$. To this end, we perform permutation tests with the Pearson's

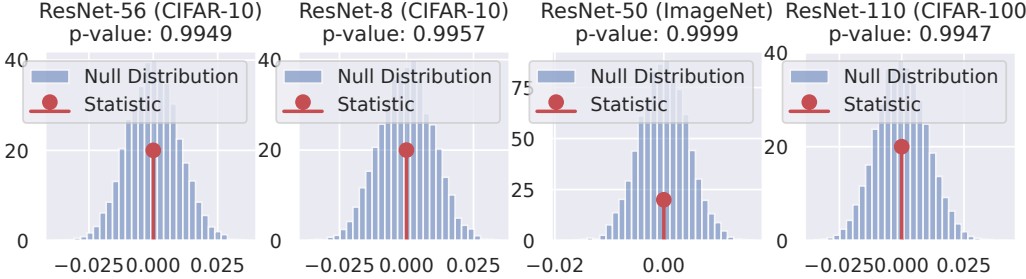

Figure 6: Null distribution, the statistic and the $p$-value of the permutation test results for the residual against $\log \mathrm{Bias}^2_{h_\theta,(x_i,y_i)}$ in linear regression of $\log \mathrm{Vari}_{h_\theta,(x_i,y_i)}$ on $\log \mathrm{Bias}_{h_\theta,(x_i,y_i)}$.

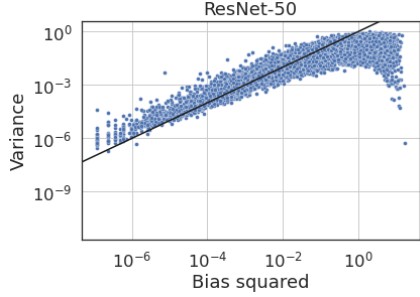

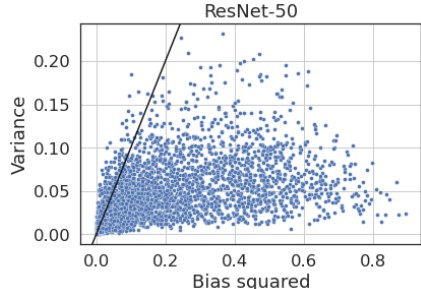

Figure 7: Sample-wise bias and variance of the CE loss.

Figure 8: Sample-wise bias and variance plotted in linear scale (with correctly classified samples only).

correlation coefficient on the residuals and their corresponding $\log \mathrm{Bias}_{h_\theta,(x_i,y_i)}$ values. The null hypothesis is that the residual and $\log \mathrm{Bias}_{h_\theta,(x_i,y_i)}$ are not correlated. We plot the null distribution, the statistic, and the $p$-value of the test results on the four models in Figure 6. The results show that the null hypothesis is not rejected, suggesting that it may be true.

### E.4 LINEAR VS LOGARITHMIC SCALE

In Section 1, the bias-variance alignment is presented first in the logarithmic scale (see Eq. (1)) and subsequently in the linear scale (see Eq. 2). Here, we provide a rigorous analysis on their connections. In addition, we explain the implication of the bias-variance alignment when plotted in the linear scale, which is complemented by empirical results.

**Connection between bias-variance alignment in linear vs logarithmic scale.** First, we provide a formal statement on the connection between the linear and log scale of the bias-variance alignment.

**Proposition E.1.** *If* $\log \mathrm{Vari}_{h_\theta,(x_i,y_i)} = \log \mathrm{Bias}^2_{h_\theta,(x_i,y_i)} + E_{h_\theta} + \varepsilon_i$ *where* $\varepsilon_i$ *is independent of* $\mathrm{Bias}^2_{h_\theta,(x_i,y_i)}$ *and* $\mathbb{E}_{i\sim\mathrm{Unif}([n])}[\varepsilon_i] = 0$, *we have*

$$\mathrm{Vari}_{h_\theta,(x_i,y_i)} = C_{h_\theta} \mathrm{Bias}^2_{h_\theta,(x_i,y_i)} + D_{h_\theta} \mathrm{Bias}^2_{h_\theta,(x_i,y_i)} \eta_i \,,$$

*where* $C_{h_\theta} = e^{E_{h_\theta}} \mathbb{E}_{i\sim\mathrm{Unif}([n])}[e^{\varepsilon_i}] > 0$, $D_{h_\theta} = e^{E_{h_\theta}} > 0$, $\eta_i = e^{\varepsilon_i} - \mathbb{E}[e^{\varepsilon_i}]$ *and* $\mathbb{E}_{i\sim\mathrm{Unif}([n])}[\eta_i] = 0$.

*Proof.* We exponentiate both sides of $\log \mathrm{Vari}_{h_\theta,(x_i,y_i)} = \log \mathrm{Bias}^2_{h_\theta,(x_i,y_i)} + E_{h_\theta} + \varepsilon_i$ where $\mathbb{E}_{i\sim\mathrm{Unif}([n])}[\varepsilon_i] = 0$ and obtain

$$\mathrm{Vari}_{h_\theta,(x_i,y_i)} = e^{E_{h_\theta}} \mathrm{Bias}^2_{h_\theta,(x_i,y_i)} e^{\varepsilon_i} = e^{E_{h_\theta}} \mathrm{Bias}^2_{h_\theta,(x_i,y_i)}(\eta_i + \mathbb{E}[e^{\varepsilon_i}])$$
$$= C_{h_\theta} \mathrm{Bias}^2_{h_\theta,(x_i,y_i)} + D_{h_\theta} \mathrm{Bias}^2_{h_\theta,(x_i,y_i)} \eta_i \,,$$

where $\eta_i$ has mean 0 by definition. $\square$

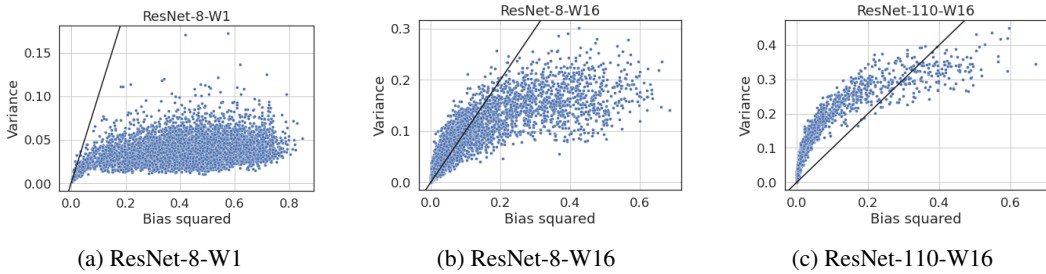

(a) ResNet-8-W1      (b) ResNet-8-W16      (c) ResNet-110-W16

Figure 9: Same as Figure 4(a-c) but plotted in linear scale and with correctly classified samples only.

**Sample-wise bias and variance plotted in linear scale.** Unlike in the log scale where the noise term $\epsilon_i$ (see Eq. (1)) is independent of the bias and variance, in linear scale the noise term $\xi_i$ is multiplied by a factor that scales with the squared bias (see Eq. 2). This implies that instead of aligning along a straight line, sample-wise bias and vairance in linear scale has a cone-shaped distribution. That is, as bias increases, an increasingly wider range of variance is covered by the samples and such a range forms a cone. To illustrate this, we regenerate the plot of Figure 1b but with linear (instead of log) scale in both the x and y axis, and the result is shown in Figure 8 (we also removed the incorrectly classified data points from the plot). Furthermore, to observe the effect of model size on bias-variance alignment in linear scale, we regenerate the Figure 4(a-c) with x and y axis switched to linear scale and present the result in Figure 9.

### E.5 CORRELATION TO PREDICTION UNCERTAINTY AND LOGIT NORM

Figure 1b demonstrates that the bias and variance of varying sample points exhibit the alignment phenomenon for points that are correctly classified. Here, in addition to the correctness of prediction, we also examine the relation between the alignment phenomenon with the prediction uncertainty and the logit norm.

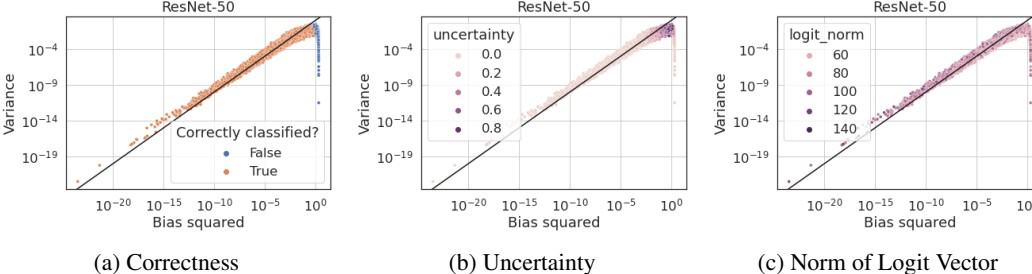

(a) Correctness      (b) Uncertainty      (c) Norm of Logit Vector

Figure 10: Same as Figure 1b, but with each sample colored according to *(a)*: Correctness of model prediction, *(b)*: Uncertainty of model prediction, and *(c)*: $\ell^2$ norm of the logit vector.

Figure 10a is the same as the one in Figure 1b for the reader's reference. In Figure 10b, we show how the uncertainty in model predictive distribution, i.e., $\text{Unce}_h(x)$ (see Definition 2.2), correlates with bias and variance. It can be seen that samples with large variance are those with large uncertainty scores. We give a formal relation between bias, variance, and uncertainty in Theorem 4.3. Finally, Figure 10c shows the lack of correlation between bias/variance and the $\ell^2$ norm of the logit vector.

### E.6 EFFECT ON THE SOURCES OF RANDOMNESS

The decomposition of the generalization into the summation of bias and variance requires one to specify a source of randomness in obtaining a collection of models. In classical bias-variance tradeoff, this source of randomness is usually taken to be the sampling of the training dataset. Correspondingly, the numerical estimation of bias and variance can be achieved by sampling a given dataset via bootstrap (see e.g. Neal et al. (2018)). This is the approach that we adopt in all numerical experiments in this paper, other than those in this section. On the other hand, modern deep networks often have other sources of randomness as well, such as the initialization of the model parameters, random sampling of the batches in the training process.

In this section, we study whether the emergence of bias-variance alignment is due purely to the randomness in sampling the training dataset, or other sources of randomness may also give rise to a similar phenomenon. Towards that, we conduct experiments to train multiple deep neural networks *without* data bootstrapping. In such cases, the randomness in the collection of networks comes only from random initialization and data batching. The result is shown in Figure 11. Comparing it with Figure 4, where the only difference lies in the bootstrapping of training dataset, it can be seen that the source of randomness have a very small impact on the bias-variance alignment phenomenon.

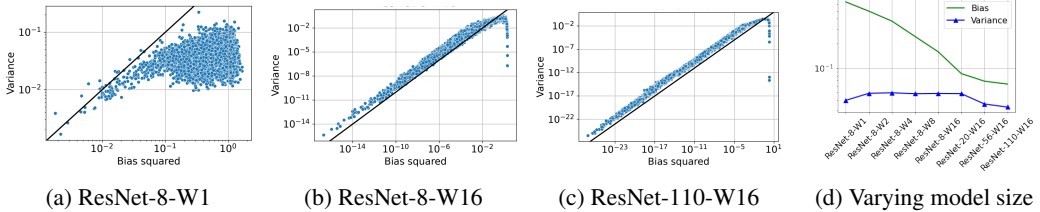

| (a) ResNet-8-W1 | (b) ResNet-8-W16 | (c) ResNet-110-W16 | (d) Varying model size |

Figure 11: Same as Figure 4 but without bootstrapping of training dataset. Hence, the randomness in computing the bias and variance comes from random initialization and data batching, and there is no randomness in sampling of training dataset.

### E.7 TREC EXPERIMENTS WITH BERT

We next show that on NLP datasets with a Transformer-based model (Vaswani et al., 2017), more specifically BERT (Devlin et al., 2019), the bias-variance alignment observation holds. This is shown in Figure 12 where we consider the TREC dataset (Hovy et al., 2001; Li & Roth, 2002) with its fine-grained labels (i.e., 47 classes) and vary the number of layers in the BERT model.

In this experiment, each of the two ensembles consists of 20 BERT models. In each case, each of these models was initialized from the same pre-trained checkpoint, and trained for 20 epochs with learning rate of $2 \times 10^{-5}$ using Adam. We use a polynomial decay learning rate schedule with the number of warm-up steps set to be 10% of the number of total update steps. Training batch size was set to 8.

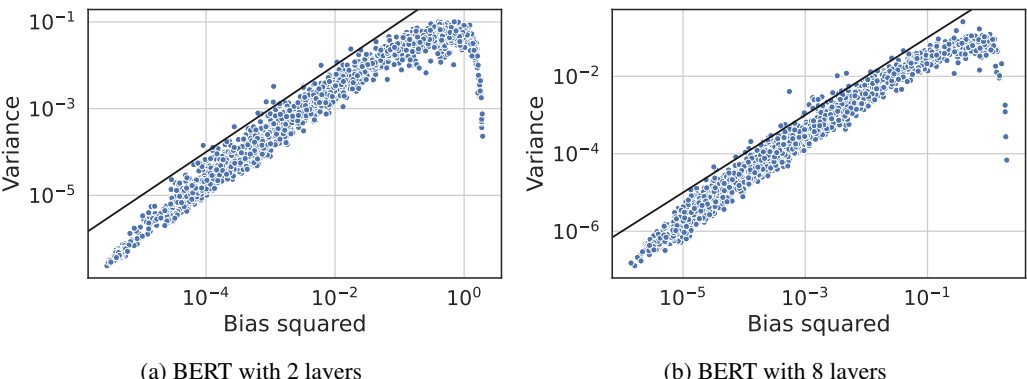

| (a) BERT with 2 layers | (b) BERT with 8 layers |

Figure 12: Sample-wise bias and variance of BERT fine-tuned on TREC. We confirm the bias-variance alignment phenomenon.

## F FURTHER RESULTS ON CALIBRATION AND THE BIAS-VARIANCE CORRELATION

### F.1 PERFECT CALIBRATION DOES NOT NECESSARILY IMPLY PERFECT CONFIDENCE CALIBRATION

Perfect calibration does not necessarily imply perfect confidence calibration. To illustrate this, consider the following example: let $\mathcal{X} = \mathcal{Y} = \{1, 2\}$, and let $X$ be a uniformly random variable on $\{1, 2\}$. Let the probability of $Y = i$ given $X$ be $\mathbb{P}(i \mid X) = \mathbf{1}\{X \neq i\}$, and let the classifier $h$ be

defined as $h(i \mid X) = \mathbf{1}\{X = i\}$. In addition, let $\Sigma_i$ represent the trivial $\sigma$-algebra for all $i$ in the set $1, 2$. In this case, we have $\mathbb{E}[\Delta(i \mid X) \mid \Sigma_i] = \mathbb{E}[\Delta(i \mid X)] = \mathbb{E}[2 \cdot \mathbf{1}\{X = i\} - 1] = 0$, which means that $h$ has perfect calibration with respect to $\{\Sigma_i\}_{i=1,2}$. However, since $\mathrm{pred}_h(x) = x$, we have $\mathbb{E}[\Delta(\mathrm{pred}_h(X) \mid X) \mid \Sigma_{\mathrm{pred}_h(X)}] = \mathbb{E}[\Delta(\mathrm{pred}_h(X) \mid X)] = \mathbb{E}[\Delta(X \mid X)] = 1$. Therefore, $h$ does not have perfect confidence calibration.

This is not true even for preimage perfect calibration $\Sigma_i^{pre}$. Consider the following counterexample

$$
h(i \mid x) = \begin{array}{c|cccc}
\diagdown i & 1 & 2 & 3 & 4 \\
x & & & & \\
\hline
1 & 0.3 & 0.25 & 0.2 & 0.25 \\
2 & 0.3 & 0.5 & 0.2 & 0
\end{array} \, ,
$$

$$
\mathbb{P}_{Y|X}(i \mid x) = \begin{array}{c|cccc}
\diagdown i & 1 & 2 & 3 & 4 \\
x & & & & \\
\hline
1 & 0.4 & 0.25 & 0.1 & 0.25 \\
2 & 0.2 & 0.5 & 0.3 & 0
\end{array} \, .
$$

We set $\mathbb{P}(X = 1) = \mathbb{P}(X = 2) = \mathrm{1/2}$. It is clear that

$$
\mathbb{E}\left[ h(i \mid X) - \mathbb{P}_{Y|X}(i \mid X) \mid h(i \mid X) \right] = 0 \tag{19}
$$

holds for for $i = 2, 4$. If $i = 1$, we have $h(i \mid X) = 0.3$. Therefore, $\mathbb{E}\left[ h(1 \mid X) - \mathbb{P}_{Y|X}(1 \mid X) \mid h(1 \mid X) = 0.3 \right] = 0.3 - \mathbb{E}\left[ \mathbb{P}_{Y|X}(1 \mid X) \mid h(1 \mid X) = 0.3 \right] = 0.3 - \frac{0.4 + 0.2}{2} = 0$. Similarly, we can show that (19) holds for $i = 4$. For $\Sigma_i^{(2)}$, $\Sigma_{\mathrm{pred}_h(X)}^{(2)} = \sigma(h(\mathrm{pred}_h(X) \mid X)) = \mathrm{conf}_h(X)$. Note that in this example, $\mathrm{conf}_h(X)$ can take only two values $0.3$ and $0.5$. Since

$$
\mathbb{E}[(\mathrm{conf}_h(X) - \mathrm{acc}_h(X)) \mid \mathrm{conf}_h(X) = 0.3] \tag{20}
$$
$$
= \mathbb{E}[(\mathrm{conf}_h(X) - \mathrm{acc}_h(X)) \mid X = 1] = -0.1 \neq 0 \, , \tag{21}
$$

perfect confidence calibration is not satisfied.

## F.2 PRE-IMAGE EXPECTED CALIBRATION ERROR

The expected calibration error (ECE) with respect to $\{\Sigma_i^{\mathrm{pre}}\}_{i \in [K]}$ recovers the ECE from Equation (2) in (Guo et al., 2017). Recall the definition of $\mathrm{conf}$ and $\mathrm{acc}$ in Section 2.1. The ECE with respect to $\{\Sigma_i^{\mathrm{pre}}\}_{i \in [K]}$ is

$$
\mathrm{ECE}_{\mathbb{P}}^{\{\Sigma_i\}_{i \in [K]}} \tag{22}
$$
$$
= \mathbb{E}\left| \mathbb{E}[\Delta(\mathrm{pred}_h(X) \mid X) \mid \Sigma_{\mathrm{pred}_h(X)}^{\mathrm{pre}}] \right| \tag{23}
$$
$$
= \mathbb{E}\left| \mathbb{E}[\Delta(\mathrm{pred}_h(X) \mid X) \mid h(\mathrm{pred}_h(X) \mid X)] \right| \tag{24}
$$
$$
= \mathbb{E}\left| \mathbb{E}[(\mathrm{conf}_h(X) - \mathrm{acc}_h(X)) \mid \mathrm{conf}_h(X)] \right| \, . \tag{25}
$$

Equation (25) follows from $\Delta(\mathrm{pred}_h(X) \mid X) = \mathrm{conf}_h(X) - \mathrm{acc}_h(X)$ and $h(\mathrm{pred}_h(X) \mid X) = \mathrm{conf}_h(X)$. This recovers the definition of the ECE in (Guo et al., 2017, Equation (2)).

## F.3 BIN-WISE EXPECTED CALIBRATION ERROR

The expected calibration error (ECE) with respect to $\{\Sigma_i^{\mathrm{bin}}\}_{i \in [K]}$ recovers the ECE from Equation (3) in (Guo et al., 2017). Let $E_j$ represent the event $\left\{ \frac{j-1}{M} < \mathrm{conf}_h(X) \leq \frac{j}{M} \right\} = \{\lceil M \, \mathrm{conf}_h(X) \rceil = j\}$.

The ECE with respect to $\{\Sigma_i^{\mathrm{bin}}\}_{i\in[K]}$ is

$$\mathrm{ECE}_{\mathbb{P}}^{\{\Sigma_i\}_{i\in[K]}} \tag{26}$$

$$= \sum_{j\in[M]} \mathbb{P}(E_j)\mathbb{E}\left[\left|\mathbb{E}[\Delta(\mathrm{pred}_h(X) \mid X) \mid \Sigma_{\mathrm{pred}_h(X)}^{\mathrm{bin}}]\right| \mid E_j\right] \tag{27}$$

$$= \sum_{j\in[M]} \mathbb{P}(E_j)\mathbb{E}\left[\left|\mathbb{E}[(\mathrm{conf}_h(X) - \mathrm{acc}_h(X)) \mid \lceil M\,\mathrm{conf}_h(X)\rceil]\right| \mid E_j\right] \tag{28}$$

$$= \sum_{j\in[M]} \mathbb{P}(E_j)\left|\mathbb{E}[(\mathrm{conf}_h(X) - \mathrm{acc}_h(X)) \mid E_j]\right|. \tag{29}$$

Equation (28) follows from $\Delta(\mathrm{pred}_h(X) \mid X) = \mathrm{conf}_h(X) - \mathrm{acc}_h(X)$ and $\lceil Mh(\mathrm{pred}_h(X) \mid X)\rceil = \lceil M\,\mathrm{conf}_h(X)\rceil$. Equation (29) is the ECE with respect to $\{\Sigma_i^{\mathrm{bin}}\}_{i\in[K]}$ on the population $\mathbb{P}(X,Y)$. If one wants to estimate the ECE from an empirical distribution formed by sampling $n$ i.i.d. samples $\{(x_i,y_i)\}_{i\in[n]}$ from $\mathbb{P}(X,Y)$, then $\mathbb{P}(E_j)$ is $\frac{|B_j|}{n}$, where $B_j$ denotes the set of the indices of the samples whose confidence falls into the bin $\left(\frac{j-1}{M}, \frac{j}{M}\right]$. Under the empirical distribution, we have

$$\mathbb{E}[\mathrm{acc}_h(X) \mid E_j] = \frac{1}{|B_j|}\sum_{i\in B_j}\mathbf{1}\{\mathrm{pred}_h(x_i) = y_i\}. \tag{30}$$

$$\mathbb{E}[\mathrm{conf}_h(X) \mid E_j] = \frac{1}{|B_j|}\sum_{i\in B_j}\mathrm{conf}_h(x_i). \tag{31}$$

We recover the definition of the ECE in (Naeini et al., 2015) and (Guo et al., 2017, Equation (3)).

### F.4 PROOF OF COROLLARY 4.5

*Proof of Corollary 4.5.* By (14) and the bounded convergence theorem, we have

$$\mathbb{E}\left[\mathbb{E}_{Y\mid X}[\beta(i)^2 - \sigma(i)^2] \mid \Sigma_i\right] \tag{32}$$

$$= \mathbb{E}\left[\mathbb{E}_\theta\left[h_\theta(i \mid X)(1 - h_\theta(i \mid X))\right] \mid \Sigma_i\right] \to 0. \tag{33}$$

Let us write $\delta(i) = \beta(i) - \sigma(i)$. It follows that

$$\mathbb{E}\left[\mathbb{E}_{Y\mid X}[\beta(i)^2 - \sigma(i)^2] \mid \Sigma_i\right] \tag{34}$$

$$= \mathbb{E}\left[\mathbb{E}_{Y\mid X}[(\sigma(i) + \delta(i))^2 - \sigma(i)^2] \mid \Sigma_i\right] \tag{35}$$

$$= \mathbb{E}\left[\mathbb{E}_{Y\mid X}[\delta(i)(\delta(i) + 2\sigma(i))] \mid \Sigma_i\right] \tag{36}$$

$$\geq \mathbb{E}\left[\mathbb{E}_{Y\mid X}[2\delta(i)^2] \mid \Sigma_i\right] \geq 0. \tag{37}$$

As a result, we obtain $\mathbb{E}\left[\mathbb{E}_{Y\mid X}[\delta(i)^2] \mid \Sigma_i\right] \to 0$, which implies $\mathbb{E}\left[\mathbb{E}_{Y\mid X}[\delta(i)] \mid \Sigma_i\right] \to 0$ since $L^2$ convergence of random variables implies $L^1$ convergence. $\square$

### F.5 PROOF OF THEOREM 4.3

*Proof of Theorem 4.3.* We have

$$\mathrm{Bias}_{h_\theta,(X,Y)}^2(i) = h(i \mid X)^2 + \mathbf{1}\{Y = i\} - 2\cdot\mathbf{1}\{Y = i\}h(i \mid X), \tag{38}$$

$$\mathrm{Vari}_{h_\theta,(X,Y)}(i) = \mathbb{E}_\theta h_\theta(i \mid X)^2 - h(i \mid X)^2. \tag{39}$$

Therefore we get

$$\mathrm{Bias}_{h_\theta,(X,Y)}^2(i) - \mathrm{Vari}_{h_\theta,(X,Y)}(i) = 2\left(h(i \mid X)^2 - \mathbf{1}\{Y = i\}h(i \mid X)\right) \tag{40}$$

$$+ \mathbf{1}\{Y = i\} - \mathbb{E}_\theta h_\theta(i \mid X)^2. \tag{41}$$

Taking the expectation over the conditional distribution of $Y \mid X$ yields

$$\mathbb{E}_{Y\mid X}\left[\mathrm{Bias}_{h_\theta,(X,Y)}^2(i) - \mathrm{Vari}_{h_\theta,(X,Y)}(i)\right]$$
$$= 2h(i \mid X)\Delta(i \mid X) + \mathbb{P}_{Y\mid X}(i \mid X) - \mathbb{E}_\theta h_\theta(i \mid X)^2. \tag{42}$$

Then we further take the conditional expectation $\mathbb{E}[\cdot \mid \Sigma_i]$ and re-arrange the terms, and obtain

$$\mathbb{E}\left[\mathbb{E}_{Y|X}[\mathrm{Bias}^2_{h_\theta,(X,Y)}(i) - \mathrm{Vari}_{h_\theta,(X,Y)}(i)] - \mathbb{P}_{Y|X}(i \mid X) + \mathbb{E}_\theta h_\theta(i \mid X)^2 \mid \Sigma_i\right]$$
$$= 2h(i \mid X)\mathbb{E}\left[\Delta(i \mid X) \mid \Sigma_i\right] . \tag{43}$$

Taking the absolute value and then the outer expectation gives

$$\mathbb{E}\left|\mathbb{E}\left[\mathbb{E}_{Y|X}[\mathrm{Bias}^2_{h_\theta,(X,Y)}(i) - \mathrm{Vari}_{h_\theta,(X,Y)}(i)] - \mathbb{P}_{Y|X}(i \mid X) + \mathbb{E}_\theta h_\theta(i \mid X)^2 \mid \Sigma_i\right]\right|$$
$$= 2\mathbb{E}\left[h(i \mid X)\left|\mathbb{E}\left[\Delta(i \mid X) \mid \Sigma_i\right]\right|\right] \tag{44}$$
$$\leq 2\mathbb{E}\left[\left|\mathbb{E}\left[\Delta(i \mid X) \mid \Sigma_i\right]\right|\right]$$

Summing (43) over $i \in [K]$ and taking the outer conditional expectation $\mathbb{E}[\cdot \mid \Sigma]$ gives

$$\mathbb{E}\left[\mathbb{E}_{Y|X}[\mathrm{Bias}^2_{h_\theta,(X,Y)} - \mathrm{Vari}_{h_\theta,(X,Y)}] \mid \Sigma\right] \tag{45}$$
$$= 1 - \mathbb{E}\left[\mathbb{E}_{Y|X}\mathbb{E}_\theta\|h_\theta(\cdot \mid X)\|_2^2 \mid \Sigma\right] + 2\sum_{i\in[K]}\mathbb{E}\left[\mathbb{E}[\Delta(i \mid X) \mid \Sigma_i]h(i \mid X) \mid \Sigma\right] . \tag{46}$$

Re-arranging the terms and taking the absolute value yields

$$\mathbb{E}\left|\mathbb{E}\left[\mathbb{E}_{Y|X}[\mathrm{Bias}^2_{h_\theta,(X,Y)} - \mathrm{Vari}_{h_\theta,(X,Y)}] - 1 + \mathbb{E}_{Y|X}\mathbb{E}_\theta\|h_\theta(\cdot \mid X)\|_2^2 \mid \Sigma\right]\right| \tag{47}$$
$$\leq 2\sum_{i\in[K]}\mathbb{E}\left[|\mathbb{E}[\Delta(i \mid X) \mid \Sigma_i]| \, h(i \mid X)\right] \leq 2\,\mathrm{CWCE}_{\mathbb{P}}^{\{\Sigma_i\}_{i\in[K]}} . \tag{48}$$

$\square$

## F.6 No bias-variance correlation in Kullback-Leibler convergence.

The expected Kullback-Leibler (KL) divergence $\mathbb{E}_\theta D_{\mathrm{KL}}(e_Y \parallel h_\theta(\cdot \mid X))$ can also be decomposed (Heskes, 1998; Zhou et al., 2021; Yang et al., 2020) into the bias $\mathrm{Bias}^2_{h_\theta,(X,Y)}$ and the variance $\mathrm{Vari}_{h_\theta,(X,Y)}$

$$\mathrm{Bias}^2_{h_\theta,(X,Y)} = D_{\mathrm{KL}}(e_Y \parallel h(\cdot \mid X)),$$
$$\mathrm{Vari}_{h_\theta,(X,Y)} = \mathbb{E}_\theta D_{\mathrm{KL}}(e_Y \parallel h_\theta(\cdot \mid X)) - \mathrm{Bias}^2_{h_\theta,(X,Y)},$$

where the mean function $h$ and the partition function $Z$ thereof are defined by

$$h(i \mid X) = \frac{1}{Z}\exp(\mathbb{E}_\theta \log h_\theta(i \mid X)), \tag{49}$$
$$Z = \sum_{i\in[K]}\exp(\mathbb{E}_\theta \log h_\theta(i \mid X)). \tag{50}$$

We can see $\mathrm{Vari}_{h_\theta,(X,Y)} = -\log Z$.

The following Proposition F.1 demonstrates that there is no correlation between bias and variance in KL divergence, unlike in mean squared error. Specifically, we prove that the ratio of expected bias to expected variance in the decomposition of KL divergence can take any value in the range of $(0, \infty)$.

**Proposition F.1.** *There exists a data distribution $\mathbb{P}(X, Y)$ such that for any value $r \in (0, \infty)$, there is an ensemble $\{h_\theta\}_\theta$ such that its mean function $\mathbb{E}_\theta h_\theta$ has samplewise perfect calibration, and the ratio of expected bias to expected variance under the KL divergence $\frac{\mathbb{E}_{Y|X}\mathrm{Bias}^2_{h_\theta,(X,Y)}}{\mathbb{E}_{Y|X}\mathrm{Vari}_{h_\theta,(X,Y)}} = r$.*

*Proof.* Suppose that there are $K = 2$ classes and for every $x$, $\mathbb{P}(i \mid x) = 1/2$ ($i = 1, 2$). Moreover, define $h_1(1 \mid x) = h_2(2 \mid x) = \varepsilon$ and $h_1(2 \mid x) = h_2(1 \mid x) = 1 - \varepsilon$, and set $\theta$ to a uniformly random variable on $\{1, 2\}$. Then the mean function $h$ satisfies $h(1 \mid x) = h(2 \mid x) = 1/2$, which does not depend on $\varepsilon$. The expected bias $\mathbb{E}_{Y|X}\mathrm{Bias}^2_{h_\theta,(X,Y)}$ is $\log 2$. The partition function $Z$ equals $2\exp((\log \varepsilon + \log(1 - \varepsilon))/2)$, from which we obtain the variance $\mathbb{E}_{Y|X}\mathrm{Vari}_{h_\theta,(X,Y)} = \mathrm{Vari}_{h_\theta,(X,Y)} = -\log 2 - \frac{1}{2}\log \varepsilon(1 - \varepsilon)$. As $\varepsilon \to 0^+$, the variance $\mathrm{Vari}_{h_\theta,(X,Y)}$ tends to $\infty$. As $\varepsilon \to 1/2$, the variance $\mathrm{Vari}_{h_\theta,(X,Y)}$ vanishes. Therefore the ratio $\frac{\mathbb{E}_{Y|X}\mathrm{Bias}^2_{h_\theta,(X,Y)}}{\mathbb{E}_{Y|X}\mathrm{Vari}_{h_\theta,(X,Y)}}$ can be any value in the range of $(0, \infty)$. $\square$

In Proposition F.1, we can let $r$ approach 0. In this limit, there exists a collection of ensembles $\{h_\theta\}_\theta$ (which depends on $r$, hence the term "collection") such that $\frac{\mathbb{E}_{Y|X} \text{Bias}^2_{h_\theta,(X,Y)}}{\mathbb{E}_{Y|X} \text{Vari}_{h_\theta,(X,Y)}} \to 0$. Conversely, we can let $r$ approach infinity, in which case $\frac{\mathbb{E}_{Y|X} \text{Bias}^2_{h_\theta,(X,Y)}}{\mathbb{E}_{Y|X} \text{Vari}_{h_\theta,(X,Y)}} \to \infty$. Therefore, either bias or variance can be arbitrarily large relative to the other, implying that there is no alignment of bias and variance under the KL divergence.

### F.7 THEOREM 4.3 IMPLIES GENERALIZATION DISAGREEMENT EQUALITY (GDE)

In this section, we show that Theorem 4.3 implies the generalization disagreement equality (GDE), which is the main result of (Jiang et al., 2022; Kirsch & Gal, 2022). We first recap the GDE using the notation of this paper. We begin with defining the test error, disagreement, and class aggregated calibration error (CACE) originally defined in (Jiang et al., 2022; Kirsch & Gal, 2022).

**Definition F.2** (Test error, disagreement, and class aggregated calibration error (Jiang et al., 2022; Kirsch & Gal, 2022)). Let $\{h_\theta : \mathcal{X} \to \mathcal{M}([K])\}$ be an ensemble of trained models, each of which has a deterministic prediction, i.e., $h_\theta(\cdot \mid x)$ is a one-hot vector for $\forall x \in \mathcal{X}$. Let $h(\cdot \mid x) \triangleq \mathbb{E}_\theta h_\theta(\cdot \mid x)$ be the mean function of $\{h_\theta\}$. Then, the test error, disagreement, and class aggregated calibration error (CACE) of $h$ are defined as follows:

$$\text{TestErr}_\mathbb{P}(h_\theta) = \mathbb{E}_{(X,Y)\sim\mathbb{P}}[\mathbf{1}\{h_\theta(\cdot \mid X) \neq e_Y\}],$$
$$\text{Dis}_\mathbb{P}(h_\theta, h_{\theta'}) = \mathbb{E}_{(X,Y)\sim\mathbb{P}}[\mathbf{1}\{h_\theta(X) \neq h_{\theta'}(X)\}],$$
$$\text{CACE}_{\mathbb{P},h} = \int_0^1 \left| \sum_{i\in[K]} \mathbb{P}(Y=i, h(i \mid X) = q) - q \sum_{i\in[K]} \mathbb{P}(h(i \mid X) = q) \right| dq.$$

Note that while the test error $\text{TestErr}_\mathbb{P}(h_\theta)$ and disagreement $\text{Dis}_\mathbb{P}(h_\theta, h_{\theta'})$ are expected values over $\mathbb{P}$, they still have randomness due to $\theta$.

Moreover, note that Jiang et al. (2022) use an integer $i \in [K]$ to denote the prediction of $h_\theta$. However, we use a one-hot vector $e_i \in \mathbb{R}^K$. We will see the mathematical convenience of representing the prediction with a one-hot vector in our proof of Theorem F.3. In particular, our proof of Theorem F.3 shows that in expectation, the disagreement is equal to the variance (defined in Equation 8) and the test error equals half the risk (defined in Equation 10).

**Theorem F.3** (Theorem 4.2 of (Jiang et al., 2022)). *If $h_\theta$ outputs an one-hot vector (as assumed in (Jiang et al., 2022)) and $\theta, \theta'$ are i.i.d., The following inequality holds:*

$$|\mathbb{E}_{\theta,\theta'}[\text{Dis}_\mathbb{P}(h_\theta, h_{\theta'})] - \mathbb{E}_\theta[\text{TestErr}_\mathbb{P}(h_\theta)]| \leq \text{CACE}_{\mathbb{P},h} \ .$$

*If the ensemble $\{h_\theta\}$ satisfies the pre-image perfect calibration ($\mathbb{E}[\Delta(i \mid X) \mid \Sigma_i^{\text{pre}}] = 0$, for $\forall i \in [K]$), the following generalization disagreement equality (GDE) holds:*

$$\mathbb{E}_{\theta,\theta'}[\text{Dis}_\mathbb{P}(h_\theta, h_{\theta'})] = \mathbb{E}_\theta[\text{TestErr}_\mathbb{P}(h_\theta)] \ .$$

*Proof.* We first show that the disagreement is equal to variance in expectation:

$$\mathbb{E}_{\theta,\theta'}[\text{Dis}_\mathbb{P}(h_\theta, h_{\theta'})] = \mathbb{E}_{\theta,\theta'}\mathbb{E}_{(X,Y)\sim\mathbb{P}}\left[ \frac{\|h_\theta(X) - h'_\theta(X)\|_2^2}{2} \right]$$

$$= \mathbb{E}_{(X,Y)\sim\mathbb{P}}\left[ \mathbb{E}_\theta[\|h_\theta(X)\|_2^2] - \|\mathbb{E}_\theta[h_\theta(X)]\|_2^2 \right] = \mathbb{E}_{(X,Y)\sim\mathbb{P}}\left[ \text{Vari}_{h_\theta,(X,Y)} \right] \ .$$

Next, we show that the test error is equal to half the risk in expectation:

$$\mathbb{E}_\theta[\text{TestErr}_\mathbb{P}(h_\theta)] = \mathbb{E}_\theta\mathbb{E}_{(X,Y)\sim\mathbb{P}}\left[ \frac{\|h_\theta(\cdot \mid X) - e_Y\|_2^2}{2} \right]$$

$$= \mathbb{E}_{(X,Y)\sim\mathbb{P}}\left[ \frac{\text{Risk}_{h_\theta,(X,Y)}}{2} \right] = \mathbb{E}_{(X,Y)\sim\mathbb{P}}\left[ \frac{\text{Bias}^2_{h_\theta,(X,Y)}}{2} + \frac{\text{Vari}_{h_\theta,(X,Y)}}{2} \right] \ .$$

We can obtain the following equation by subtracting the above two equations:

$$\mathbb{E}_{\theta,\theta'}[\text{Dis}_{\mathbb{P}}(h_\theta, h_{\theta'})] - \mathbb{E}_\theta[\text{TestErr}_{\mathbb{P}}(h_\theta)]$$

$$= \mathbb{E}_{(X,Y)\sim\mathbb{P}} \left[ \frac{\text{Bias}^2_{h_\theta,(X,Y)}}{2} + \frac{\text{Vari}_{h_\theta,(X,Y)}}{2} - \text{Vari}_{h_\theta,(X,Y)} \right].$$

$$= \mathbb{E}_{(X,Y)\sim\mathbb{P}} \left[ \frac{\text{BVG}_{h_\theta,(X,Y)}}{2} \right]$$

Apply Theorem 4.3 with $\Sigma_i = \Sigma_i^{\text{pre}} = \sigma(h(i \mid X))$ and using $\text{Unce}_{h_\theta}(X) = 0$ (because $h_\theta(\cdot \mid x)$ is a one-hot vector for $\forall x \in \mathcal{X}$), we obtain

$$\mathbb{E}_{(X,Y)\sim\mathbb{P}} \left[ \frac{\text{BVG}_{h_\theta,(X,Y)}}{2} \right] = \sum_{i\in[K]} \mathbb{E}\left[\mathbb{E}[\Delta(i \mid X) \mid \Sigma_i^{\text{pre}}]h(i \mid X)\right]. \tag{51}$$

We see immediately that if $\mathbb{E}[\Delta(i \mid X) \mid \Sigma_i^{\text{pre}}] = 0$ for $\forall i \in [K]$, the GDE is satisfied.

The right-hand side of Equation (51) equals

$$\sum_{i\in[K]} \mathbb{E}\left[\mathbb{E}[\Delta(i \mid X) \mid \Sigma_i^{\text{pre}}]h(i \mid X)\right]$$

$$= \int_0^1 \sum_{i\in[K]} \left(q - \mathbb{P}(Y = i \mid h(i \mid X) = q)\right) q\mathbb{P}(h(i \mid X) = q)dq$$

$$= \int_0^1 \left(q \sum_{i\in[K]} \mathbb{P}(h(i \mid X) = q) - \sum_{i\in[K]} \mathbb{P}(Y = i, h(i \mid X) = q)\right) qdq.$$

In the first equality, we expand the expectations. To compute the outer expectation, we condition on the prediction $h(i|X)$ returned by the model for class index $i$ and integrate with respect to the conditional probability distribution $h(i|X)$. The inner expectation is taken over all $X$ such that the model for class index $i$ returns $q$, the value that we condition on. In the last equality, we apply the conditional probability rule $\mathbb{P}(Y = i, h(i \mid X) = q) = \mathbb{P}(Y = i \mid h(i \mid X) = q)\mathbb{P}(h(i \mid X) = q)$.

Taking the absolute value gives

$$|\mathbb{E}_{\theta,\theta'}[\text{Dis}_{\mathbb{P}}(h_\theta, h_{\theta'})] - \mathbb{E}_\theta[\text{TestErr}_{\mathbb{P}}(h_\theta)]|$$

$$\leq \int_0^1 \left|\left(q \sum_{i\in[K]} \mathbb{P}(h(i \mid X) = q) - \sum_{i\in[K]} \mathbb{P}(Y = i, h(i \mid X) = q)\right)\right| qdq$$

$$\leq \int_0^1 \left|\left(q \sum_{i\in[K]} \mathbb{P}(h(i \mid X) = q) - \sum_{i\in[K]} \mathbb{P}(Y = i, h(i \mid X) = q)\right)\right| dq$$

$$= \text{CACE}_{\mathbb{P},h},$$

where the last inequality uses $q \in [0, 1]$. $\qquad\square$

## G  FURTHER RESULTS ON NEURAL COLLAPSE AND THE BIAS-VARIANCE CORRELATION

### G.1  PROPERTIES OF SIMPLEX EQUIANGULAR TIGHT FRAME (ETF)

$W^{\text{ETF}}$ has the following properties: First, it is symmetric. Second, the inner product between any two distinct columns is equal to $-\frac{1}{K-1}$. Third, this pairwise distance is maximized, i.e., there does not exist any matrix where the inner product between any two distinct pairs of columns are smaller than $-\frac{1}{K-1}$.

## G.2   VERIFYING ASSUMPTION 5.1

As stated in Assumption 5.1, we assume that the logits of a deep network for any test sample $(X, Y)$ are drawn from the following distribution:

$$
\begin{aligned}
&\text{softmax}\left(W^{\text{ETF}}(sw_Y^{\text{ETF}} + v)\right) \\
&= \text{softmax}\left(\sqrt{\frac{K}{K-1}}\left(I_K - \frac{1}{K}\mathbf{1}_K\mathbf{1}_K^\top\right)\left(s\sqrt{\frac{K}{K-1}}\left(e_Y - \frac{1}{K}\mathbf{1}_K\right) + v\right)\right) \\
&\qquad \left(\text{use the fact }\left(I_K - \frac{1}{K}\mathbf{1}_K\mathbf{1}_K^\top\right)\mathbf{1}_K = 0\right) \\
&= \text{softmax}\left(\sqrt{\frac{K}{K-1}}\left(I_K - \frac{1}{K}\mathbf{1}_K\mathbf{1}_K^\top\right)\left(s\sqrt{\frac{K}{K-1}}e_Y + v\right)\right) \\
&= \text{softmax}\left(\frac{sK}{K-1}e_Y + \sqrt{\frac{K}{K-1}}v - \sqrt{\frac{K}{K-1}}\cdot\frac{1}{K}\mathbf{1}_K\mathbf{1}_K^\top\left(s\sqrt{\frac{K}{K-1}}e_Y + v\right)\right) \\
&= \text{softmax}\left(\frac{sK}{K-1}e_Y + \sqrt{\frac{K}{K-1}}v\right),
\end{aligned}
\tag{52}
$$

where the final equality follows from the fact that the term $-\sqrt{\frac{K}{K-1}}\cdot\frac{1}{K}\mathbf{1}_K\mathbf{1}_K^\top\left(s\sqrt{\frac{K}{K-1}}e_Y + v\right)$ is parallel to $\mathbf{1}_K$ and the addition of this term does not change the value of the softmax function. In particular, $v$ above is a random vector with i.i.d. entries drawn according to $-\beta\sqrt{\frac{K}{K-1}}v_i \sim \text{Gumbel}(\mu, \beta)$. In this section, we verify this assumption from two perspectives. First, we will plot the distributions of logits in practical neural networks, and show that they align with (52). Second, we will show through simulation that, if the logits are generated according to (52), then we observe bias-variance alignment.

**Distribution of logits.** From (52), the logits corresponding to the correct class (i.e., $Y$) and any incorrect class $Y' \neq Y$ are given by

$$
\frac{sK}{K-1} + \sqrt{\frac{K}{K-1}}v_Y, \quad \text{and} \quad \sqrt{\frac{K}{K-1}}v_{Y'},
\tag{53}
$$

respectively. In particular, since Gumbel distribution has a unimodal shaped probability density function, the distribution of both the positive and all the negative logits have unimodal shape according to (53). To verify that this aligns with the practical observations, we calculate the distributions of logit values on various datasets and model architectures, for both positive classes and negative classes. The results are presented in Figure 13. We observe unimodal logit distributions for both positive and negative classes in all cases. On the other hand, one may notice that while (53) predicts the positive and negative logits to have different biases but the same variance, in many cases from Figure 13, the positive and negative logits have notable different variances. Hence, Assumption 5.1 is used as a simplified model that makes our theoretical analysis tractable, but is not meant to perfectly model the distribution of logits in practice. We will show next that such a simplified model is sufficient for producing the bias-variance alignment phenomenon that we observe in practice.

**Synthesizing bias-variance alignment.** To justify Assumption 5.1, we synthetically generate a collection of logit vectors according to (53), and plot the sample-wise bias and variance obtained from the logit vectors. Specifically, given any number $n$ as the number of samples, and $K$ as the number of classes, we first generate a collection of $n$ random labels where each label is drawn uniformly at random from $[K]$. For each label, we sample $T$ logit vectors independently according to (53) (for the Gumbel distribution, we take $\mu = 0$ and $\beta = 1$). Here, $T$ is interpreted as the number of independently trained models for estimating bias and variance.

The results with $n = 200, K = 2$, and $T = 10$, under varying choices of $s \in \{5, 10, 20, 100\}$ are reported in Figure 14. In all cases, we observe a clear bias-variance alignment. In particular, we did not draw correctly and incorrectly classified samples in different colors (as in Figure 1b) because all the samples in these cases are correctly classified.

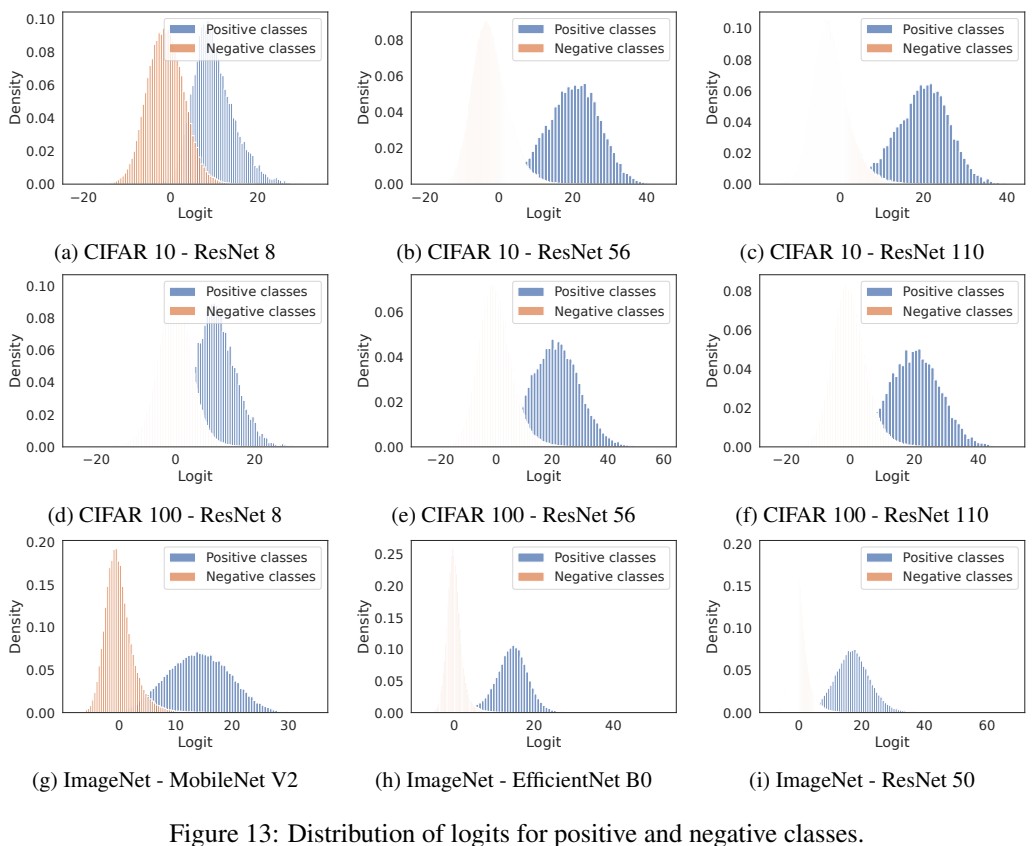

Figure 13: Distribution of logits for positive and negative classes.

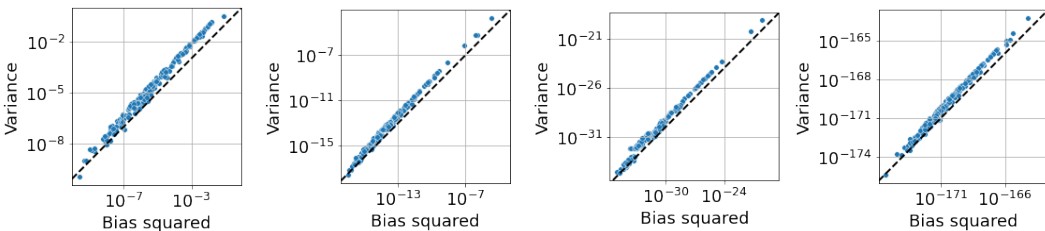

Figure 14: Sample-wise bias and variance for synthetic data generated according to (53). From left to right, $s$ is varied in the set $\{5, 10, 20, 100\}$.

### G.3 VERIFYING COROLLARY 5.3: BINARY CLASSIFICATION

We note that the Neural collapse theory relies on the binary classification assumption. To ensure that the bias-variance alignment results hold for such a setup empirically, we construct a binary classification problem based on the CIFAR-10 dataset: each example in the first five classes is assigned label 0, and each eample in the last five classes is assigned label 1. We call the resulting dataset CIFAR-2. The results are shown in Figure 15.

### G.4 RELATIONSHIP BETWEEN GUMBEL AND EXPONENTIAL DISTRIBUTION

**Lemma G.1.** *Let* $X \sim \text{Gumbel}(\mu, \beta)$. *Then,* $e^{-X/\beta} \sim \text{Exp}(e^{\mu/\beta})$.

*Proof.* Recall that the cumulative distribution function (CDF) of the Gumbel distribution is given by

$$\mathbb{P}(X \leq x) = e^{-e^{-(x-\mu)/\beta}}. \tag{54}$$

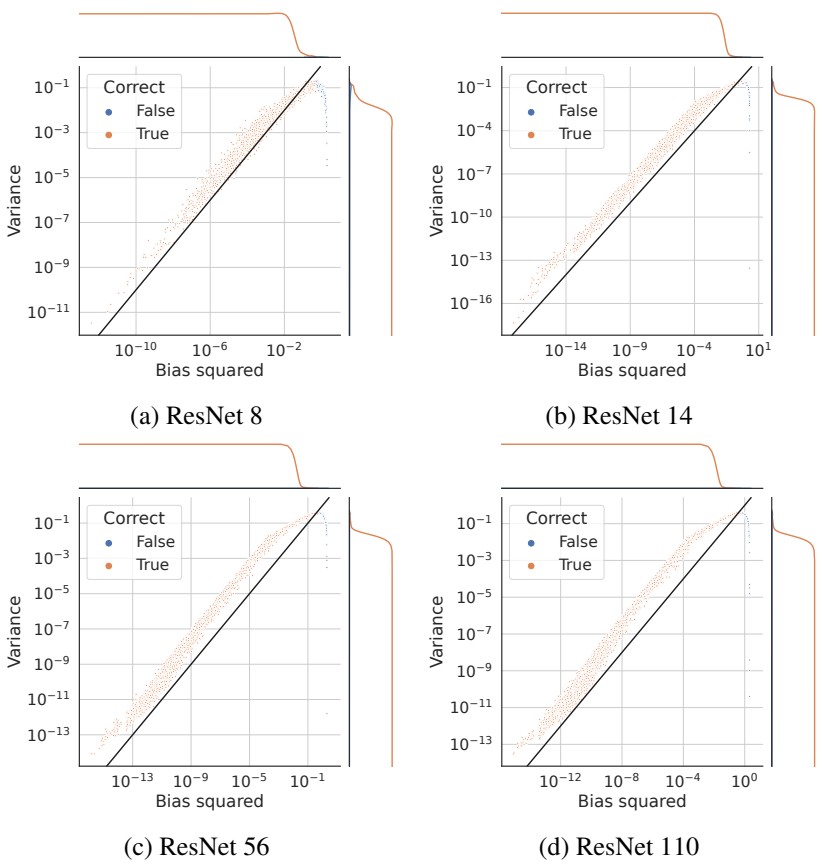

Figure 15: Squared bias and variance computed based on various model sizes on the CIFAR-2 problem. See Appendix G.3 for details.

Thus we get

$$\mathbb{P}(e^{-X/\beta} \leq e^{-x/\beta}) = 1 - e^{-e^{-(x-\mu)/\beta}} . \tag{55}$$

Substituting $t = e^{-x/\beta}$, we get

$$\mathbb{P}(e^{-X/\beta} \leq t) = 1 - e^{-te^{\mu/\beta}} . \tag{56}$$

This is the CDF of $\mathrm{Exp}(e^{\mu/\beta})$. $\qquad\square$

### G.5 PROOF OF THEOREM 5.2

*Proof of Theorem 5.2.* As the first step, we compute the output of function $h_\theta$

$$
\begin{aligned}
h_\theta(X) &= \mathrm{softmax}\left(W\psi_\tau(X)\right) \\
&= \mathrm{softmax}\left(W^{\mathrm{ETF}}R^\top\left(R\left(sw_Y^{\mathrm{ETF}} + v\right)\right)\right) \\
&= \mathrm{softmax}\left(\sqrt{\frac{K}{K-1}}\left(s\sqrt{\frac{K}{K-1}}\left(e_Y - \frac{1}{K}\mathbf{1}_K\right) + v\right)\right) \\
&= \mathrm{softmax}\left(\frac{sK}{K-1}e_Y + \sqrt{\frac{K}{K-1}}v\right)
\end{aligned}
\tag{57}
$$

Let us denote $w \triangleq h_\theta(\cdot \mid X)$. Without loss of generality and for the ease of presentation, we label the $Y$-th entry as the first entry ($Y = 1$). Moreover, we introduce the shorthand notation $u \triangleq \sqrt{\frac{K}{K-1}}v$.

Then we have

$$s = \text{softmax}\left(\frac{sK}{K'}e_1 + u\right)$$

$$= \left(\frac{ae^{u_1}}{ae^{u_1} + e^{u_2} + \cdots + e^{u_K}}, \frac{e^{u_2}}{ae^{u_1} + e^{u_2} + \cdots + e^{u_K}}, \ldots, \frac{e^{u_K}}{ae^{u_1} + e^{u_2} + \cdots + e^{u_K}}\right)^{\top} \tag{58}$$

where $a = e^{sK/K'}$. In light of Lemma G.1, since $-\beta u_i \sim \text{Gumbel}(\mu, \beta)$ are i.i.d., then $v_i \triangleq e^{u_i} \sim \text{Exp}(\lambda)$ where $\lambda \triangleq e^{\mu/\beta}$.

Let us look at the first entry $w_1$ of $s = \text{softmax}(re_1 + u)$. It equals

$$w_1 = \frac{ae^{u_1}}{ae^{u_1} + e^{u_2} + \cdots + e^{u_K}} = \frac{a}{a + \frac{e^{u_2} + \cdots + e^{u_K}}{e^{u_1}}} = \frac{a}{a + (K-1)F} = \frac{c}{c+F}, \tag{59}$$

where $c = \frac{a}{K-1} = \frac{e^{sK/K'}}{K'}$ and $F = \frac{(e^{u_2} + \cdots + e^{u_K})/(2(K-1))}{e^{u_1}/2} = \frac{(e^{u_2} + \cdots + e^{u_K})/(K-1)}{e^{u_1}} \sim \text{F}(2(K-1), 2)$ follows the F distribution (this is because $e^{u_i} \sim \text{Exp}(\lambda) = \frac{1}{2\lambda}\chi_2^2$).

The expectation of $w_1/c$ is given by

$$\mathbb{E}\left[\frac{w_1}{c}\right] = \mathbb{E}_{F \sim \text{F}(2K', 2)}\left[\frac{1}{c+F}\right] \tag{60}$$

$$= \frac{1}{K' \text{Beta}(K', 1)} \int_0^{\infty} \frac{x^{K'-1}}{(c+x)(x+1/K')^{K'+1}} dx \tag{61}$$

$$= \frac{c^{K'-1}\left(c - \frac{1}{K'}\right)^{-K'}(cK' - K'\log(cK') - 1)}{cK' - 1} \tag{62}$$

$$+ \frac{\left(c - \frac{1}{K'}\right)^{-K'-1}}{K'} \sum_{j=1}^{K'-1} \frac{(K'-j)(c - \frac{1}{K'})^j c^{-j+K'-1}}{j} \tag{63}$$

$$= \phi_{K'}(c). \tag{64}$$

As a result, the squared bias of the first entry $w_1$ is

$$\beta_{h_\theta,(X,Y)}(1) = |\mathbb{E}w_1 - 1| = |c\phi_{K'}(c) - 1|. \tag{65}$$

To get the variance of the first entry $w_1$, we compute its second moment as the first step:

$$\mathbb{E}[w_1^2] = c^2 \mathbb{E}\left[\frac{1}{(c+F)^2}\right] = -c^2 \frac{d}{dc}\mathbb{E}\left[\frac{1}{c+F}\right] = -c^2 \frac{d\phi_{K'}(c)}{dc}. \tag{66}$$

Therefore, it follows that

$$\text{Vari}_{h_\theta,(X,Y)}(1) = \mathbb{E}[w_1^2] - \mathbb{E}[w_1]^2 = -c^2 \frac{d\phi_{K'}(c)}{dc} - c^2 \phi_{K'}(c)^2, \tag{67}$$

which yields

$$\varsigma_{h_\theta,(X,Y)}(1) = \sqrt{\text{Vari}_{h_\theta,(X,Y)}(1)} = c\sqrt{-\left(\frac{d\phi_{K'}(c)}{dc} + \phi_{K'}(c)^2\right)}. \tag{68}$$

$\square$

### G.6 PROOF OF COROLLARY 5.3

*Proof of Corollary 5.3.* If $K = 2$, we get $\mathbb{E}_{F \sim \text{F}(2,2)}\left[\frac{1}{c+F}\right] = \frac{c - \log(c) - 1}{(c-1)^2}$. As in the Proof of Theorem 5.2, without loss of generality and for the ease of presentation, we label the $Y$-th entry as the

first entry ($Y = 1$). We then have the following expectations:

$$\mathbb{E}_u[w_1] = \frac{c(c - \log(c) - 1)}{(c-1)^2}, \quad \mathbb{E}_u[w_2] = \frac{-c + c\log(c) + 1}{(c-1)^2}$$

$$\mathbb{E}_{F \sim \mathrm{F}(2,2)}\left[\frac{1}{(c+F)^2}\right] = -\frac{\partial}{\partial c}\mathbb{E}_{F \sim \mathrm{F}(2,2)}\left[\frac{1}{c+F}\right] = \frac{c^2 - 2c\log(c) - 1}{(c-1)^3 c}$$

To obtain the variance of $w_1, w_2$, we first calculate the second moment:

$$\mathbb{E}_u[w_1^2] = \frac{c^2(c^2 - 2c\log(c) - 1)}{(c-1)^3 c} \tag{69}$$

$$\mathbb{E}_u[w_2^2] = \mathbb{E}_u[(1 - w_1)^2] = \frac{c^2 - 2c\log(c) - 1}{(c-1)^3} \tag{70}$$

As a result, we have

$$\mathrm{Var}_u[w_1] = \mathrm{Var}_u[w_2] = \frac{c\left((c-1)^2 - c\log^2(c)\right)}{(c-1)^4} \tag{71}$$

Therefore, we obtain

$$\beta_{h_\theta,(X,Y)}(1) = |\mathbb{E}_u[w_1] - 1| = \frac{|\log(c)c - c + 1|}{(c-1)^2},$$

$$\beta_{h_\theta,(X,Y)}(2) = |\mathbb{E}_u[w_2] - 0| = \beta_{h_\theta,(X,Y)}(1)$$

$$\varsigma_{h_\theta,(X,Y)}(1) = \varsigma_{h_\theta,(X,Y)}(2) = \sqrt{\mathrm{Var}_u[w_1]} = \frac{\sqrt{c\left((c-1)^2 - c\log^2(c)\right)}}{(c-1)^2}.$$

The ratio $\frac{\beta_{h_\theta,(X,Y)}(i)}{\varsigma_{h_\theta,(X,Y)}(i)} = \frac{|-c+c\log(c)+1|}{\sqrt{c((c-1)^2 - c\log^2(c))}}$ is a decreasing function of $c \in (1, \infty)$ and $\lim_{c \to 1+} \frac{\beta_{h_\theta,(X,Y)}(i)}{\varsigma_{h_\theta,(X,Y)}(i)} = \sqrt{3}$. Therefore, the ratio $\frac{\beta_{h_\theta,(X,Y)}(i)}{\varsigma_{h_\theta,(X,Y)}(i)} \leq \sqrt{3} < 1.74$ for $c > 1$. To show $\frac{\beta_{h_\theta,(X,Y)}(i)}{\varsigma_{h_\theta,(X,Y)}(i)} = \frac{|-c+c\log(c)+1|}{\sqrt{c((c-1)^2 - c\log^2(c))}} \geq \frac{\log c - 1}{\sqrt{c}} \equiv \frac{2s-1}{e^s}$, it suffices to prove

$$(-c + c\log(c) + 1)^2 \geq \left((c-1)^2 - c\log^2(c)\right)(\log(c) - 1)^2,$$

which is equivalent to

$$\log(c)\left(-2c + c\log^3(c) - 2c\log^2(c) + (3c - 1)\log(c) + 2\right) \triangleq \log(c)f(c) \geq 0.$$

Since $f'(c) = -\frac{1}{c} + \log^3(c) + \log^2(c) - \log(c) + 1 \geq 0$ and $f(1) = 0$, we complete the proof for $\frac{\beta_{h_\theta,(X,Y)}(i)}{\varsigma_{h_\theta,(X,Y)}(i)} \geq \frac{\log c - 1}{\sqrt{c}}$. On the log scale, $\frac{\log \mathrm{Bias}_{h_\theta,(X,Y)}(i)}{\log \mathrm{Vari}_{h_\theta,(X,Y)}(i)} = \frac{\log\left(\frac{(-c+c\log(c)+1)^2}{(c-1)^4}\right)}{\log\left(\frac{c((c-1)^2 - c\log^2(c))}{(c-1)^4}\right)}$ is a monotone function for $\forall c > 1$. As $c$ approaches 1 from the right, the limit of the function is $\frac{\log(4)}{\log(12)} > 0.557$. As $c$ approaches infinity, the limit of the function is 2. $\qquad\square$

## H  FUTURE WORK

In the realm of future research, an intriguing avenue lies in the exploration of the bias-variance alignment phenomenon within diverse machine learning model ensembles. While our present work sheds light on this phenomenon in the context of deep neural network ensembles, there remains a substantial gap in our understanding of its presence or absence in other model types such as decision trees and support vector machines. An essential direction for future investigation involves unraveling the root causes of bias-variance alignment, particularly as it pertains to distinct types of overfitting in machine learning models—be it benign, tempered, or catastrophic (Mallinar et al., 2022). Our conjecture extends towards discerning whether bias-variance alignment is a ubiquitous trait across

various overfitting scenarios or if its manifestation is contingent on the specific type of overfitting. Notably, we hypothesize that catastrophic overfitting may exhibit a bias-variance tradeoff, while benign and tempered overfitting might demonstrate a unique bias-variance alignment. Unraveling these intricacies is pivotal for advancing our comprehension of ensemble behaviors across different machine learning paradigms. Moreover, Atanasov et al. (2022) demonstrates an inverse relationship between the feature learning strength and the variance across initializations. Exploring the interplay between bias, variance, and feature learning strength in neural networks is also an intriguing area of future research.

