# OpenReview forum: "On Bias-Variance Alignment in Deep Models"
_ICLR.cc/2024/Conference — ICLR 2024 spotlight_

### Official Review · Reviewer_1psh · 2023-10-31

**Soundness:** 4 excellent
**Presentation:** 4 excellent
**Contribution:** 4 excellent
**Rating:** 8
**Confidence:** 4

**Summary:**

The paper conducts extensive experiments across realistic datasets and architectures (mostly on vision tasks) to demonstrate that the bias of a classifier correlates strongly with its variance in the regime where the number of parameters is large. They go on to show that smaller networks deviate and have worse bias-variance correlation. They prove theoretically that such an alignment should happen under the assumption that the network is well-calibrated. Finally, they show that the theory of neural collapse also predicts such an alignment.

**Strengths:**

The paper provides extensive empirical evidence for the bias-variance alignment phenomenon across different model architectures and datasets. The figures are very compelling, esp Figures 2-4. The two theoretical motivations are solid in the opinion of the reviewer. The connection to calibration is certainly interesting. Overall, this paper provides new insights into the bias-variance relation in deep learning and I recommend acceptance.

**Weaknesses:**

I don't have many comments on this. The primary datasets that this paper focuses on are in image classification - it would be good to see results on other domains like NLP or even simple polynomial curve fitting.

It would also be good to be more explicit about what bias and variance you're talking about. Adlam and Pennington (as cited in our paper) show that double descent can be clearly understood in linear models by using a "fine-grained" bias-variance decomposition. There, there is variance due to initial parameters, train set, and label noise. Its extremely hard to understand which sources of variance you're talking about in section 2.1. It's clear that initial parameters is certainly one of them, since you are defining $h(\cdot | x) = \mathbb E_{\theta} h_{\theta}(\cdot | x)$, but it is not clear how the variance over train set is entering this (or whether it is at all).

The change to CE loss in appendices E.2, F.6 is not very readable. It seems that the effect mostly goes away. Given the ubiquity of CE in modern ML, it would be nice to expand and make these currently tiny sections more clear.

**Questions:**

My understanding is that real-world models do not generally exhibit calibration. Given this, would you still expect the bias-variance equality to hold approximately for uncalibrated models as well?

I don't really understand why you're saying that in Figure 7, Bias and Variance don't align well. There definitely seems to be nontrivial correlation. Is it just that it is worse than Figures 3, 4?

It would be interesting to study the bias and variance as a function of feature learning strength in the network. The scale of the variance over initializations is shown to depend inversely on feature learning strength in e.g. https://arxiv.org/abs/2212.12147

---

> ### Author Response · Authors · 2023-11-21
> **Response to Reviewer 1psh**
>
> Thank you so much for your time, insightful feedback, and positive evaluation :-)
>
>
> > Q1: I don't have many comments on this. The primary datasets that this paper focuses on are in image classification - it would be good to see results on other domains like NLP or even simple polynomial curve fitting.
>
>
> A: Following the reviewer’s suggestion we conducted an empirical study of sample-wise bias and variance in polynomial curve fitting. We observe that, when learning a polynomial with a higher order than that of the groundtruth polynomial, the variance linearly correlates with the squared bias across all test samples. This observation is akin to the bias-variance alignment reported in our paper for neural networks, in the sense that a linear correlation emerges in both cases. However, a notable difference is that the linear correlation in polynomial fitting is observed in the linear scale, as opposed to the log scale as in neural networks. We plan to conduct a dedicated study of this connection and difference in future work.
>
>
> > Q2: It would also be good to be more explicit about what bias and variance you're talking about. Adlam and Pennington (as cited in our paper) show that double descent can be clearly understood in linear models by using a "fine-grained" bias-variance decomposition. There, there is variance due to initial parameters, train set, and label noise. Its extremely hard to understand which sources of variance you're talking about in section 2.1. It's clear that initial parameters is certainly one of them, since you are defining $h(\cdot | x) = \mathbb E_{\theta} h_{\theta}(\cdot | x)$, but it is not clear how the variance over train set is entering this (or whether it is at all).
>
>
> A: Thank you for your insightful question and feedback. We appreciate the opportunity to clarify our approach in Section 2.1. In Section 2.1, we aim to establish a comprehensive theoretical framework that defines the ensemble, its bias, variance, and related concepts. The sources of randomness, including the initial parameters and the training set, can partly or collectively contribute to the overall variability in our models in the empirical results. In accordance with your comment, we have added a footnote in Section 2.1 to elaborate on this.
>
>
> It is important to note that the specific source of randomness, whether it be initial parameters, train set, or label noise, is not explicitly isolated in the definition. Instead, we treat them as part of the broader randomness inherent in the resulting trained parameters denoted by $\theta$. Thus, the ensemble is represented by ${h_\theta}$, and $\theta$ captures the randomness introduced by these sources.
>
>
> Furthermore, we would like to draw your attention to Appendix E.6, titled "Effect On The Sources Of Randomness," where we conducted empirical analyses to assess the impact of various sources of randomness on our model.
>
>
> We hope that these clarifications address your concerns. We value your feedback and are committed to refining our manuscript to enhance its clarity and precision. Thank you once again for your thoughtful engagement with our work.

---

> ### Author Response · Authors · 2023-11-21
> **Response to Reviewer 1psh (continued #1)**
>
> > Q3: The change to CE loss in appendices E.2, F.6 is not very readable. It seems that the effect mostly goes away. Given the ubiquity of CE in modern ML, it would be nice to expand and make these currently tiny sections more clear.
>
>
> A: We appreciate your feedback regarding the readability of the change to CE loss in appendices E.2 and F.6. With regard to your question in the second sentence, please see our response to Q5 as well. In response, we have expanded these appendices to provide a more comprehensive explanation of the bias-variance decomposition under the KL divergence and the implications of Proposition F.1.
>
>
> In Appendix E.2, we have added a new paragraph that provides an intuitive explanation of the bias-variance decomposition. The paragraph states that $\bar{h}(\cdot|X)$ represents the average prediction under the KL divergence, assigning a probability proportional to $\exp E_\theta \log(h_\theta(i \mid X))$ to each class i. The bias term measures the KL divergence between the true distribution $e_Y$​ and the average prediction $\bar{h}(\cdot|X)$, quantifying the deviation of the ensemble's mean prediction from the actual class distribution. The variance term, on the other hand, captures the average KL divergence between the individual predictions in the ensemble and the mean prediction $\bar{h}(\cdot|X)$, reflecting the overall variability of the ensemble's predictions.
>
>
> In Appendix F.6, we have added a new paragraph that discusses the implications of Proposition F.1. The paragraph states that in Proposition F.1, we can let r approach 0. In this limit, there exists a collection of ensembles {hθ​}θ​ (which depends on r, hence the term "collection") such that $\frac{ E_{Y\mid X} Bias}{ E_{Y\mid X}  Vari} \to 0$. Conversely, we can let r approach infinity, in which case $\frac{ E_{Y\mid X} Bias}{ E_{Y\mid X}  Vari} \to \infty$. Therefore, either bias or variance can be arbitrarily large relative to the other, implying that there is no alignment of bias and variance under the KL divergence.
>
>
> We believe that these additions make the appendices more clear, informative, and easier to understand.
>
>
> > Q4: My understanding is that real-world models do not generally exhibit calibration. Given this, would you still expect the bias-variance equality to hold approximately for uncalibrated models as well?
>
>
> A: Thank you for your question. You are correct that real-world models generally do not exhibit calibration. However, our extensive numerical results (for example, Figure 1(b) and Figure 3), across different neural architectures and different datasets demonstrate that the bias-variance equality approximately holds in practice, even for uncalibrated models. This observation suggests that the bias-variance alignment is a fundamental property of neural networks, and that it does not require the calibration assumption to strictly hold.
>
>
> > Q5: I don't really understand why you're saying that in Figure 7, Bias and Variance don't align well. There definitely seems to be nontrivial correlation. Is it just that it is worse than Figures 3, 4?
>
>
> A: Thank you for your question regarding Figure 7, and we appreciate the opportunity to provide further clarification on this matter.
>
>
> In our submission, we explicitly mentioned, "In Figure 7, we present the sample-wise bias and variance from decomposing the CE loss" (last paragraph of Appendix E.2). It's important to note that Figure 7 utilizes the bias and variance decomposed from the cross-entropy (CE) loss, as opposed to the mean squared error used in Figures 3 and 4.
>
>
> As stated in the submission, "It can be seen that the bias no longer aligns well with the variance" (last paragraph of Appendix E.2). The key point here is that, when decomposing the CE loss, the alignment between bias and variance is not as pronounced as observed in Figures 3 and 4.
>
>
> Additionally, we provided a theoretical result in Proposition F.1, illustrating that there is no bias-variance correlation under the Kullback–Leibler divergence. This theoretical underpinning supports our observation in Figure 7.
>
>
> We hope this clarification addresses your question. If you have any further questions or if there are additional aspects you would like us to elaborate on, please feel free to let us know.

---

> ### Author Response · Authors · 2023-11-21
> **Response to Reviewer 1psh (continued #2)**
>
> > Q6: It would be interesting to study the bias and variance as a function of feature learning strength in the network. The scale of the variance over initializations is shown to depend inversely on feature learning strength in e.g. https://arxiv.org/abs/2212.12147
>
>
> A: Thank you for your valuable suggestion regarding the study of bias and variance as a function of feature learning strength in the network. We appreciate your insightful input and find it particularly relevant to our work.
>
>
> Upon your recommendation, we explored the paper referenced in your question (https://arxiv.org/abs/2212.12147), and we agree that it provides valuable insights into the scale of variance over initializations, a point highlighted by you. In our revised manuscript, we have cited this paper and incorporated a discussion of its findings in the new "Future Work" section (Appendix H).
>
>
> Your suggestion has enriched our exploration of potential avenues for future research, and we are grateful for your contribution. If you have any further recommendations or if there are additional aspects you believe would enhance the scope of our work, please feel free to share them.

---

### Official Review · Reviewer_iQea · 2023-10-31

**Soundness:** 2 fair
**Presentation:** 2 fair
**Contribution:** 3 good
**Rating:** 6
**Confidence:** 3

**Summary:**

This paper investigates the bias-variance decomposition of deep ensembles. The paper claims that, in the deep learning regime, the relationship between (squared) bias and variance is (a) approximately linear for correctly classified test examples and (b) bounded for all examples ($B^2 \geq k \cdot V$). These claims are evaluated on standard deep learning image benchmarks. The role of over-parameterization is also investigated. Two theoretical discussions are included: one showing that certain relationships between bias and variance can be made explicit under calibration assumptions and the other providing some results from the perspective of neural collapse theory.

**Strengths:**

* This paper addresses an interesting problem which is the bias-variance trade-off in the regime of deep ensembles which has some distinct characteristics compared to the regime from which it emerged (decision trees, random forests, etc).
* It makes an interesting and (to the best of my knowledge) novel claim that bias and variance are _aligned_ (i.e. have some consistent behaviors in their relationship) in the deep ensemble setting.
* Experiments are performed on standard image benchmarks of CIFAR and ImageNet making them relevant to many previous deep ensemble works.
* Aspects of the connection to calibration were useful and provided concrete results that seem to reinforce the authors claims.
* I found that the introduction did a good job of setting up the structure of the paper and it's main claims.

**Weaknesses:**

I am quite borderline on this paper. I found the topic and claims of the paper interesting and worthwhile and I certainly think there is value to the community in this type of analysis. However, the exposition of many of the ideas could have been better. I think there were too many ideas (often quite disjoint) that were densely squeezed into the 9 pages requiring a terse, notation-heavy delivery that was challenging to follow at times. I thought some ideas were under-explored (e.g. the role of overparameterization) while others were over-explored (e.g. I felt the section on neural collapse added little to the paper). Overall, I think the paper would greatly benefit from prioritizing its more impactful claims and resolving them more comprehensively. I list my more specific issues below.
* **Title** - I'm not sure about the use of the word alignment. You wish to make a statement about the relationship between two variables (i.e. they are linear or proportional). I don't think aligned quite means this. Additionally, this is an overloaded word with connotations in e.g. AI safety.
* **Related work** - I found the related work to be _highly_ incomplete without any inclusion of previous works in the ensemble or deep ensemble literature. For example, many previous works (some recent [1,2], others older [3]) have explored various overlapping aspects of decomposing ensembles into bias and variance. Recent works have empirically noticed and explored the role of overparameterization on variance [4]. Other work has noticed that optimizing to increase ensemble variance causes a simultaneous increase in the predictive bias [5]. I would recommend that the authors perform an extensive literature review to appropriately position themselves relative to this and other works from the ensemble literature.
* **Evidence for empirical claims** - Ultimately the two main hypothesized relationships proposed in this paper are empirical in nature (despite being possibly motivated by theory). Therefore I would have expected a more extensive empirical evaluation of their claims. Only CIFAR and ImageNet datasets are included, which seems insufficient for the claim that this is a general relationship. I would be more convinced if either (a) the claims were reduced to just standard benchmarks on image data or (b) more comprehensive experiments were performed (i.e. more datasets and more architectures). Additionally, I found it challenging at times to tell if the results included had sufficient detail to evaluate the claims. For example, should Fig 3 not include colors so that we can ensure the outliers are also misclassified? Should the actual bound not be included so that we can verify eqn (3)? I think the presented results could be more clearly linked to the claims they are investigating.
* **Overparameterization** - I think the role of overparameterization in deep ensembles and how it seems to invalidate the theory developed in the original ensembling literature (e.g. random forests) is fascinating. Therefore I was disappointed that this analysis didn't go into more depth and consisted of simply evaluating ResNets of various widths on CIFAR. Given that the authors decided to include this in the paper I would have liked to have learned more than what was already established in works such as [4].
* **Calibration** - The relationship between calibration and bias-variance was interesting to explore. However, given that the main claims of the paper are empirical I thought this section could have been less detailed to provide more time to actually verify and evaluate the empirical claims in depth. I thought much of this section could have been presented in a more clear way that more succinctly linked to its purpose (i.e. its relationship to bias-variance alignment and upper-bounded variance).  I think it would be possible to restructure this section such that important points are clear with some more of the technical detail (when less relevant to the main point) placed in the appendix.
* **Neural collapse** - In contrast, I thought the section on Neural collapse added very little to the paper. The assumptions are restrictive and the results don't seem to be particularly useful. Also, it seems that the section is too short to have space the provide sufficient detail and background on the topic. I would suggest moving most or all of this section to the appendix and using the additional space to expand on the empirical evaluation.


Other points
* The two claims on page 2 are empirical claims but are presented in a way that suggests they are theoretical truths (I misunderstood this on my first read). I would suggest making this more clear in the text.
* Eqn (10) should either contain a proof (in the appendix) or cite a source (since the authors claim it is well-known).
* Many of the plots are very small (especially the text) making them hard to read.
* Tables 1 & 4 formatting is unusual. Why the double line at the top? I would try to avoid vertical lines in the body of tables (see NeurIPS style guide).
* Typo - p9, l5: "In above" -> "In the above"


[1] Gupta, N., Smith, J., Adlam, B., & Mariet, Z. E. (2022). Ensembles of Classifiers: a Bias-Variance Perspective. Transactions on Machine Learning Research.

[2] Ortega, L. A., Cabañas, R., & Masegosa, A. (2022, May). Diversity and generalization in neural network ensembles. In International Conference on Artificial Intelligence and Statistics (pp. 11720-11743). PMLR.

[3] Brown, G., Wyatt, J. L., Tino, P., & Bengio, Y. (2005). Managing diversity in regression ensembles. Journal of machine learning research, 6(9).

[4] Abe, T., Buchanan, E. K., Pleiss, G., & Cunningham, J. P. (2023). Pathologies of Predictive Diversity in Deep Ensembles. arXiv preprint arXiv:2302.00704.

[5] Jeffares, A., Liu, T., Crabbé, J., & van der Schaar, M. (2023). Joint Training of Deep Ensembles Fails Due to Learner Collusion. arXiv preprint arXiv:2301.11323.

**Questions:**

* Is the $C_{h_{\theta}}$ in eqn (2) the same as the one in eqn (3)? If so, this should be made more clear.
* I would be interested in the authors' view on the consequences of the findings of this paper and how it should impact future work. For example, how do these results impact the intuitions we may have developed in the domain of ensembles of "simple" models (e.g. decision trees)?

---

> ### Author Response · Authors · 2023-11-21
> **Response to Reviewer iQea**
>
> Thank you for taking the time to read and review our paper!
>
>
> > Q: Title - I'm not sure about the use of the word alignment. You wish to make a statement about the relationship between two variables (i.e. they are linear or proportional). I don't think aligned quite means this. Additionally, this is an overloaded word with connotations in e.g. AI safety.
>
>
> A: Thank you for your careful review and insightful comment. We appreciate your feedback regarding the use of the term "bias-variance alignment" in the title of our paper.
>
> We acknowledge that the word "alignment" has multiple meanings in different contexts, and we understand your concern that it might not be the most precise term to convey the relationship between the bias and the variance in our work. However, we carefully considered several alternative terms, including "linear relationship," "proportional relationship," and "correlation," but ultimately decided to retain "bias-variance alignment" for the following reasons:
>
> 1. **Clarity and Conciseness:** The term "bias-variance alignment" is concise and easily understood, effectively conveying the idea that bias and variance are related in a specific way, namely that they are not independent of each other but rather exhibit a relationship.
>
> 2. **Contrast with Bias-Variance Trade-off:** The term "bias-variance alignment" serves to contrast with the more commonly used term "bias-variance trade-off." While the trade-off suggests that bias and variance are inversely related, our work highlights that they can also be aligned, implying a more nuanced relationship.
>
> 3. **Contextual Understanding:** We believe that readers familiar with machine learning will understand the intended meaning of "bias-variance alignment" without confusion with other uses of the term "alignment" in AI safety or other areas.
>
> We understand that there is always room for debate about the choice of words in scientific writing, and we are open to considering alternative phrasing in future work. However, we believe that the term "bias-variance alignment" effectively conveys the essence of our findings and provides a clear distinction from the traditional bias-variance trade-off perspective.
>
> Thank you again for your valuable feedback. We appreciate your thoughtful consideration of our work.

---

> ### Author Response · Authors · 2023-11-21
> **Response to Reviewer iQea (continued #1)**
>
> > Q: Related work - I found the related work to be highly incomplete without any inclusion of previous works in the ensemble or deep ensemble literature. For example, many previous works (some recent [1,2], others older [3]) have explored various overlapping aspects of decomposing ensembles into bias and variance. Recent works have empirically noticed and explored the role of overparameterization on variance [4]. Other work has noticed that optimizing to increase ensemble variance causes a simultaneous increase in the predictive bias [5]. I would recommend that the authors perform an extensive literature review to appropriately position themselves relative to this and other works from the ensemble literature.
>
>
> A: Thank you for these valuable references. As part of our rebuttal revision, we have included a comprehensive "Further Related Work" section (Appendix C) that cites and discusses the relevant papers you mentioned. For each of these papers, we have provided an in-depth discussion to highlight their contributions.
>
> In the final version of our manuscript, we plan to integrate a condensed version of this related work discussion into the main body, ensuring that the reader receives a concise yet thorough overview of the existing literature.
>
> It is important to note, however, that none of these works uncovers our key finding of the per-sample bias-variance alignment. Your insights have been instrumental in refining our work, and we appreciate your thoughtful review.
>
> The discussion in the "Further Related Work" section is as follows:
>
> "Gupta et al. (2022) investigate the theoretical underpinnings of ensemble methods for classification tasks, extending the bias-variance decomposition to derive generalized laws of total expectation and variance for nonsymmetric losses. Their work sheds light on the mechanisms by which ensembles reduce variance and potentially bias, providing valuable insights for improving the performance of ensemble classifiers.  Ortega et al. (2022) provide a comprehensive theoretical framework for understanding the relationship between diversity and generalization error in neural network ensembles. They  analyze  the  impact  of  diversity  on  ensemble  performance  for  various  loss  functions  and model combination strategies, offering valuable insights for designing effective ensemble learning algorithms. Brown et al. (2005) focus on managing diversity in regression ensembles. They introduce a control mechanism through the error function, demonstrating its effectiveness in improving ensemble performance over traditional methods.  This work provides insights into systematic control of the bias-variance-covariance trade-off in regression ensembles. Abe et al. (2023) challenge conventional wisdom on predictive diversity in deep neural network ensembles.  While diversity benefits small models, the authors find that encouraging diversity harms high-capacity deep ensembles used for classification. Their experiments show that diversity-encouraging regularizers hinder performance, suggesting that the best strategy for deep ensembles may involve using more accurate but less diverse component models. In contrast to traditional ensemble methods, deep ensembles of neural networks offer the potential for direct optimization of overall performance. However, Jeffares et al. (2023) reveal that jointly minimizing ensemble loss induces base learners to collude, inflating artificial diversity. This pseudo-diversity fails to generalize, highlighting limitations in direct joint optimization and its impact on generalization gaps."
>
> > Q: Additionally, I found it challenging at times to tell if the results included had sufficient detail to evaluate the claims. For example, should Fig 3 not include colors so that we can ensure the outliers are also misclassified?
>
> A: In response to your query, we direct your attention to Figure 10 in Appendix E.5, which comprises three subfigures. Our evaluation not only considers correctness (Figure 10(a)) but also delves into uncertainty (Figure 10(b)) and the norm of the logit vector (Figure 10(c)). Taking into account your valuable suggestion, we have incorporated color differentiations in the updated Figure 3 to enhance clarity between correctly and incorrectly classified points.

---

> ### Author Response · Authors · 2023-11-21
> **Response to Reviewer iQea (continued #2)**
>
> > Q: Should the actual bound not be included so that we can verify eqn (3)? I think the presented results could be more clearly linked to the claims they are investigating.
>
> A: . In response to your suggestion, we would like to highlight that we have already incorporated a visual aid in the form of a black reference line in Figure 3. This line corresponds to $C_{h_\theta}$ = 1, indicating that the variance is less than or equal to the square of the bias (i.e., variance <= bias^2). The positioning of correctly classified points around this reference line serves as an illustration of our findings. Additionally, as per your suggestion, in the revised Figure 3, we explicitly differentiate between correctly and incorrectly classified points.
>
> > Q: Overparameterization - I think the role of overparameterization in deep ensembles and how it seems to invalidate the theory developed in the original ensembling literature (e.g. random forests) is fascinating. Therefore I was disappointed that this analysis didn't go into more depth and consisted of simply evaluating ResNets of various widths on CIFAR. Given that the authors decided to include this in the paper I would have liked to have learned more than what was already established in works such as [4].
>
> A: Thank you for referencing the work by Abe et al. (2023). We have now added a detailed discussion about it to the rebuttal revision. Their findings emphasize the potential detrimental effects of encouraging diversity within deep neural network ensembles for classification tasks. Specifically, their experiments suggest that prioritizing accuracy over diversity in component models may yield better overall performance.
> In response to your feedback, we want to clarify that our paper offers a more nuanced and refined quantitative analysis of the interplay between bias (a metric for ensemble performance) and variance (a metric for ensemble prediction diversity). We extend beyond the established works, such as (Abe et al., 2023), by proposing a quantitative relationship indicating that bias is approximately equal to variance on a per-sample level. We not only state that high variance corresponds to high bias but also provide quantitative evidence and analysis supporting this claim. Furthermore, we introduce and prove, under the calibration assumption, the equality between bias and variance.
>
> In response to your inquiry regarding the bias and variance of the ensembles of random forests, we have included a dedicated section entitled "Future Work" in the revised manuscript (Appendix H) to address this question specifically. Our ongoing research is delving into whether the bias-variance alignment phenomenon observed in the context of deep neural network ensembles extends to other machine learning model ensembles, including decision trees, support vector machines, and others.
>
> As we explore this extension, we are actively investigating the root causes behind the observed bias-variance alignment. In our conjecture, we posit a connection between this phenomenon and various types of overfitting in machine learning models. Specifically, we reference the work of Mallinar et al. (2022) in our exploration of different overfitting types: benign, tempered, and catastrophic.
>
> Drawing inspiration from this refined taxonomy of overfitting, we are currently exploring whether bias-variance alignment is a common trait across machine learning models exhibiting each type of overfitting. Our preliminary conjecture is that catastrophic overfitting may manifest a bias-variance tradeoff, while benign and tempered overfitting could potentially exhibit bias-variance alignment.
>
> This avenue of research represents a promising direction for extending the implications of our current findings and contributes to a nuanced understanding of the interplay between overfitting types and bias-variance behaviors in ensemble models. We are actively working on gathering empirical evidence to support these conjectures and intend to provide a more comprehensive analysis in future iterations of our research.
>
> Once again, we appreciate your thoughtful question, which points at important directions for extending our work.
>
> Mallinar, N., Simon, J., Abedsoltan, A., Pandit, P., Belkin, M., & Nakkiran, P. (2022). Benign, tempered, or catastrophic: Toward a refined taxonomy of overfitting. Advances in Neural Information Processing Systems, 35, 1182-1195.

---

> ### Author Response · Authors · 2023-11-21
> **Response to Reviewer iQea (continued #3)**
>
> > Q: Calibration - The relationship between calibration and bias-variance was interesting to explore. However, given that the main claims of the paper are empirical I thought this section could have been less detailed to provide more time to actually verify and evaluate the empirical claims in depth. I thought much of this section could have been presented in a more clear way that more succinctly linked to its purpose (i.e. its relationship to bias-variance alignment and upper-bounded variance). I think it would be possible to restructure this section such that important points are clear with some more of the technical detail (when less relevant to the main point) placed in the appendix.
>
> A: Thank you for your thoughtful feedback on the calibration section of our paper. We appreciate your suggestion to streamline the presentation of technical details. In response to your comments, we would like to highlight that the submitted version has already placed most of the technical intricacies, including all proofs, in the appendix.
>
> In the "Calibration Implies Bias-Variance Alignment" section, we have deliberately focused on presenting the main theorem (Theorem 4.3) and its two corollaries (Corollary 4.4 and Corollary 4.5). The main theorem (Theorem 4.3) plays a pivotal role in characterizing the bias-variance gap and establishing its relationship with the calibration error. Corollary 4.4 is a key result that demonstrates the bias-variance alignment under the calibration assumption, showing that if an ensemble is perfectly calibrated and exhibits high confidence, then the bias equals the variance. This corollary is crucial in substantiating our main claim regarding the bias-variance alignment.
>
> Furthermore, Corollary 4.5 delves deeper into the subject, providing additional insights and demonstrating that, under the calibration assumption, bias-variance alignment occurs on an entrywise basis. These three technical results are the cornerstone of our work, and we believe their inclusion in the main body is essential for clarity and coherence.
>
> We acknowledge your suggestion to enhance clarity further and will carefully consider your feedback while revising the section. However, preserving these core results in the main body is integral to ensuring that our empirical claims are firmly grounded in the underlying theoretical framework. We hope this clarifies the rationale behind the current structure of the calibration section.
>
> > Q: Neural collapse - In contrast, I thought the section on Neural collapse added very little to the paper. The assumptions are restrictive and the results don't seem to be particularly useful. Also, it seems that the section is too short to have space the provide sufficient detail and background on the topic. I would suggest moving most or all of this section to the appendix and using the additional space to expand on the empirical evaluation.
>
> A: We appreciate your feedback on the section discussing neural collapse in our paper. While we understand your perspective on the limitations of the neural collapse section, it's important to note that other reviewers have expressed positive views on this section.
>
> For instance, Reviewer ehNn highlighted the strength of our neural collapse section in their review, stating, "The authors found two quite simple theoretical scenarios that can replicate the bias-variance alignment. Moreover, these scenarios are quite distinct: the calibration part requires no specific assumptions and provides rigorous upper bounds, while the neural collapse part uses a very specific model which admits an exact and explicit solution (i.e. not just an upper/lower bound)." This positive assessment underscores the value of the distinct theoretical scenarios we present, showcasing the diversity of our approach.
>
> Additionally, in the "Strengths" section of Reviewer 18mF's review, it is noted that "Assumption 5.1 on model prediction under neural collapse is very interesting. Using this to show the bound the ratio of variance and bias is novel." This acknowledgment reinforces the novelty and interest surrounding Assumption 5.1 and its role in establishing bounds on the ratio of variance and bias under neural collapse.
>
> While we understand your suggestion to consider moving this section to the appendix to make room for expanding the empirical evaluation, the positive feedback from other reviewers indicates that the neural collapse section has its merits. We will carefully consider your input and aim to strike a balance between addressing your concerns and preserving the valuable contributions of the neural collapse analysis in our paper.

---

> ### Author Response · Authors · 2023-11-21
> **Response to Reviewer iQea (continued #4)**
>
> > Q: The two claims on page 2 are empirical claims but are presented in a way that suggests they are theoretical truths (I misunderstood this on my first read). I would suggest making this more clear in the text.
>
>
> A: Thank you for your insightful feedback. We would like to highlight that these claims are indeed supported by our underlying theory, specifically under the calibration assumption, as demonstrated in Section 4, with Corollary 4.4 providing further corroboration. Moreover, we would like to draw your attention to Table 1, where we have taken care to clearly and explicitly indicate which results are theoretical and which are empirical.
>
> Nevertheless, we acknowledge that additional clarification in the main text can help clarify the nature of our contributions. In response to your suggestion, we have revised the introduction of the two claims on page 2 to underscore their empirical nature. The modified sentence now reads:
>
> "Our key **empirical** observations can be summarized with the following two statements, which we call the Bias-Variance Alignment and the Upper Bounded Variance."
>
> We hope that our revision addresses your concern effectively, and we hope that it contributes to a clearer understanding of our manuscript.
>
>
>
> > Q: Eqn (10) should either contain a proof (in the appendix) or cite a source (since the authors claim it is well-known).
>
> A: Thank you for your careful review and insightful comment. Indeed, Eq. (10) is the standard bias-variance decomposition and is widely used in statistics and machine learning. It is well-established and can be found in numerous textbooks and references. For instance, it is presented in Eq. (2.25) of the book "The Elements of Statistical Learning: Data Mining, Inference, and Prediction" by Hastie, Tibshirani, and Friedman (2009). We have added a reference to this book in the rebuttal revision.
>
> > Q: Many of the plots are very small (especially the text) making them hard to read.
>
> A: We appreciate your feedback regarding the size of the plots and the text within them. We will address this issue by remaking the figures and making them larger and easier to read. We will also increase the size of the text within the figures so that it is more legible. We will make these changes in the final version of the manuscript.
>
>
>
> > Q: Is the $C_{h_{\theta}}$ in eqn (2) the same as the one in eqn (3)? If so, this should be made more clear.
>
>
> A: Yes, the $C_{h_{\theta}}$ in Equation (2) is the same constant as in Equation (3). We have added a sentence to the manuscript to make this more clear.
>
> > Q: I would be interested in the authors' view on the consequences of the findings of this paper and how it should impact future work. For example, how do these results impact the intuitions we may have developed in the domain of ensembles of "simple" models (e.g. decision trees)?
>
> A: Thank you for your insightful question and for engaging with our work. We appreciate your interest in the consequences of the findings presented in our paper and how they might influence future work in the domain of ensembles, particularly those comprised of "simple" models such as decision trees.
>
> In response to your query, we have incorporated a dedicated "Future Work" section in the revised manuscript (Appendix H) to address precisely this line of inquiry. Our ongoing research delves into whether the bias-variance alignment phenomenon observed in the context of deep neural network ensembles extends to other machine learning model ensembles, including decision trees, support vector machines, and others.
>
> As we explore this extension, we are actively investigating the root causes behind the observed bias-variance alignment. In our conjecture, we posit a connection between this phenomenon and various types of overfitting in machine learning models. Specifically, we reference the work of Mallinar et al. (2022) in our exploration of different overfitting types: benign, tempered, and catastrophic.
>
> Drawing inspiration from this refined taxonomy of overfitting, we are currently exploring whether the bias-variance alignment is a common trait across machine learning models exhibiting each type of overfitting. Our preliminary conjecture is that catastrophic overfitting may manifest a bias-variance tradeoff, while benign and tempered overfitting could potentially exhibit bias-variance alignment.
>
> This avenue of research represents a promising direction for extending the implications of our current findings and contributes to a nuanced understanding of the interplay between overfitting types and bias-variance behaviors in ensemble models. We are actively working on gathering empirical evidence to support these conjectures and intend to provide a more comprehensive analysis in future iterations of our research.
>
> Mallinar, N., Simon, J., Abedsoltan, A., Pandit, P., Belkin, M., & Nakkiran, P. (2022). Benign, tempered, or catastrophic: Toward a refined taxonomy of overfitting. NeurIPS, 35, 1182-1195.

---

> > ### Comment · Reviewer_iQea · 2023-11-21
> >
> > I am very grateful to the reviewers for their comprehensive response. I respond to a selection of points below:
> >
> > > Title
> >
> > While I still maintain that an alternative title would be preferable, the current title is not problematic and I would not reject on this basis.
> >
> > > Related work
> >
> > I am happy that the authors have extended this discussion. For a final version, I highly recommend (a) extending this to include works beyond what I thought of in my review and (b) updating the current draft once the authors have had a chance to read these works in detail.
> >
> > > Overparmeterization and future work
> >
> > I thank the reviewers for their response to these points. Although I would still have appreciated more depth on the interplay between model capacity and bias-variance alignment, I accept if the authors would prefer to leave that for future work instead.
> >
> > > Calibration and Neural Collapse
> >
> > Thank you for the response. Considering that others in the community may disagree with me on this point, I am happy to accept the author's rebuttal.
> >
> > > Eqn 10
> >
> > I believe there is some confusion on this decomposition. The version you are including seems to be around the error which typically also should include an irreducible noise term. The citation you provide is an exception because they assume a setting in which the DGP is deterministic. This is not an assumption you have made (or should make). On the other hand, an exact decomposition holds when decomposing the parameters in terms of bias and variance, but this is not what you intended I don't believe.
> >
> > See the following link for further details on this distinction: https://stats.stackexchange.com/questions/320352/relation-between-mse-and-bias-variance
> >
> > > Presentation improvements
> >
> > I appreciate the authors' promised improvements to the paper's presentation. I would highly recommend spending some time making additional improvements for a camera ready to maximize the clarity and impact of the work.
> >
> > **Summary**
> >
> > Although I do retain some of my initial concerns, I am sufficiently convinced that this paper provides a useful contribution to the literature and is worthy of acceptance at this time. Therefore I have increased my score.

---

> > > ### Author Response · Authors · 2023-11-22
> > > **Thank you!**
> > >
> > > Thank you very much for reading the rebuttal and raising your score!

---

### Official Review · Reviewer_18mF · 2023-11-01

**Soundness:** 3 good
**Presentation:** 3 good
**Contribution:** 3 good
**Rating:** 6
**Confidence:** 3

**Summary:**

The paper studies the trends in the bias and variance of an ensemble of classification models (including their pointwise trends). It empirically establishes two phenomena: the bias variance alignment and upper bounding variance with bias. On a theoretical front, the paper shows that the bias variance gap in expectation can be bounded by a variant of calibration error and therefore when the models are calibrated the exhibit the above phenomenon. Furthermore, assuming that for test data the model prediction is perturbed from the model obtained during training after neural collapse and the upper bounding variance phenomenon for binary classification is proved.

**Strengths:**

a) Through experiments, the paper establishes the alignment of the bias and variance. The role of overparameterization in the bias variance phenomenon is well demonstrated. This provides an avenue to understand generalization for over-parameterized models.

b) The Assumption 5.1 on model prediction under neural collapse is very interesting. Using this to show the bound the ratio of variance and bias is novel.

c) A general definition of calibration under a sub-$\sigma$-algebras is a nice contribution.

**Weaknesses:**

a) The main weakness of the paper is the comparison with related work. For example, theorem 4.3 seems equivalent to the theorem 4.2 in [1].  In the appendix it is claimed that the theorem 4.3 implies the result of [1].  However, it seems more of an equivalence than implication.  The authors should comment on this aspect in the main paper to be more transparent.

[1] Y. Jiang, V. Nagarajan, C. Baek, and Z. Kolter. Assessing generalization of SGD
via disagreement. In International Conference on Learning Representations, 2022.

b) Assuming that the model predictions are well calibrated seems is a very strong assumption, thus using this to bound the bias variance gap makes the result less interesting.

**Questions:**

-

---

> ### Author Response · Authors · 2023-11-21
> **Response to Reviewer 18mF**
>
> Thank you so much for your time, insightful feedback, and positive evaluation :-)
>
>
> > Q: a) The main weakness of the paper is the comparison with related work. For example, theorem 4.3 seems equivalent to the theorem 4.2 in [1]. In the appendix it is claimed that the theorem 4.3 implies the result of [1]. However, it seems more of an equivalence than implication. The authors should comment on this aspect in the main paper to be more transparent.
>
>
> > [1] Y. Jiang, V. Nagarajan, C. Baek, and Z. Kolter. Assessing generalization of SGD via disagreement. In International Conference on Learning Representations, 2022.
>
> A: We appreciate the reviewer's feedback on the comparison with related work, particularly regarding Theorem 4.3 and its relation to Theorem 4.2 in [1].
> We want to clarify that our Theorem 4.3 is not equivalent to Theorem 4.2 in [1], although there are similarities. Here are the key differences:
> 1) Scope of the Results:
> - Theorem 4.2 in [1] focuses on the **entire dataset**, providing an inequality relating disagreement and test error across all data points.
> - Our Theorem 4.3 focuses on a **per-example** analysis, providing both an equality and an inequality for the gap between **per-example** bias and variance. This provides a more fine-grained understanding of the relationship between the bias and variance.
> 2) Tightness of the Inequality:
> - Our Theorem 4.3 offers a strictly tighter inequality than Theorem 4.2 in [1]. In our proof of Theorem 4.2 in [1] (Appendix F.7) starting from our Theorem 4.3, the last inequality leverages the fact that $q\le 1$ to arrive at Theorem 4.2 in [1].This inequality cannot be reversed, and thus the bound becomes less tight when arriving at the result of [1].
>
> > Q: b) Assuming that the model predictions are well calibrated seems is a very strong assumption, thus using this to bound the bias variance gap makes the result less interesting.
>
>
> A: Thank you for your feedback. We'd like to address a potential misunderstanding regarding our approach. We want to clarify that we do not make the assumption that model predictions are inherently well-calibrated. Rather, our methodology involves bounding the bias-variance gap by considering the calibration error of an ensemble. This approach does not rely on assuming good calibration of individual models; instead, we establish a connection between the bias-variance gap and the level of calibration. If you have additional questions or concerns, please feel free to reach out.

---

### Official Review · Reviewer_ehNn · 2023-11-03

**Soundness:** 4 excellent
**Presentation:** 4 excellent
**Contribution:** 4 excellent
**Rating:** 8
**Confidence:** 3

**Summary:**

The paper discusses the *bias-variance alignment* phenomenon with respect to an ensemble of deep models in classification tasks. Essentially, the phenomenon states that the logarithms of bias and variance (within the ensemble and at a given sample) of predicted probabilities are well described by a linear relation with coefficient $1$, i.e. $\log \mathrm{Vari} \approx \log \mathrm{Bias}^2 + C$. The authors first empirically demonstrate the phenomenon on image classification tasks. Then, they proceed to a theoretical justification of the phenomenon from two independent perspectives: calibration and neural collapse. For the calibration perspective, the authors propose a definition of calibration that unifies other definitions previously used in the literature and then utilize it to show that bias-variance gap can be bounded by the calibration error. For the neural collapse perspective, the authors propose a simple model of ensemble predictions based on Gumbel noise added to the perfect neural collapse prediction (i.e. ETF complex vertices) and explicitly derive bias and variance given the scale $s$ of last layer features, subsequently showing that for all values of $s$ bias-variance alignment approximately holds.

**Strengths:**

Overall, the paper is well-written and organized, which allowed to compactly describe a number of various contributions.

The contributions of the paper present are very balanced and include
- Formulation and explanation of a new phenomenon. It is done via a nice illustration (Figure 1, both a. and b.), intuitive explanation of the phenomenon, clear definitions and the setting (sec. 2.1), and a focused related work section.
- Very convincing empirical evidence supporting it. In my experience, it is quite difficult to find simple relations that accurately describe modern neural networks on realistic-data (i.e. not toy models). Yet, the quality of the linear relation on figure 1.b (and other plots of the same type) is surprisingly high.
- The authors found two quite simple theoretical scenarios that can replicate the bias-variance alignment. Moreover, these scenarios are quite distinct: the calibration part requires no specific assumptions and provides rigorous upper bounds, while the neural collapse part uses very specific model which admits an exact and explicit solution (i.e. not just an upper/lower bound).

**Weaknesses:**

Overall, I have not found significant drawbacks.

There are a few moments which could be corrected to improve clarity and reading experience:
- Formally, eq. (52) is correct only up to the addition of a vector parallel to $\mathbf{1}_K$. Indeed, such vectors, when passed to softmax, can be ignored and therefore does not affect the subsequent calculations performed in the paper. However, it would be better to mention this to avoid confusion.
- The colors for negative and positive classes in figures 12,13 are swapped compared to the rest of the plots in the papers. Though completely stylistic, it would be nice to have a consistent color scheme to make paper reading more comfortable.
- It took me some time to verify that variable $F$ appearing in eq. (59) indeed has $F(2(K-1),2)$ distribution. Since this fact is central for the following computation and, to my understanding, motivates the choice of Gumbel distribution for $v$ in assumption 5.1, it would be better to discuss and explain this explicitly in appendix.

Also, a couple of minor typos:
- Histograms for negative classes have disappeared in the middle row of Figure 12.
- A typo "$K$ as the number of samples" at the end of sec. G.2.

**Questions:**

I have a few questions mostly related to the Neural Collapse part.
- Does figure 13 contain only correctly classified samples or all of them? In the latter case, it would be interesting to know why the approximately vertical shape of wrongly classified points (e.g. a "blue tail" on figure 1.b) is not present in the synthetic data.
- *Choice of Gumbel distribution*. Overall, the validation of assumption 5.1 presented in sec. G.2 seems to support the hypothesis that we may write $\psi_\tau(X)=R(sw^{\mathrm{ETF}_Y+v})$ with random vector $v$ whose entries are i.i.d., zero mean, and have unimodal shape. For example, Gaussian distribution satisfies these criteria. So is there some additional evidence for choosing Gumbel distribution, besides it enabling an analytical solution?
- After eq. (10) it is mentioned that the paper considers (except sec. E.2) MSE loss on top of classifier probabilities. However, I did not find any mention that this is also applied to the training of neural networks used for empirical results - that would be quite non-standard from a practical point of view (CE loss is usually used in practice). Could you please provide details on how the networks were trained?
- If, in the previous question, CE loss was used, I would expect that scale parameter $s$ of ETF complex grows with training time in the terminal phase of training (it is required for reduction of train loss to extremely small values). Then, by training the network for a very long time, one can hope to cover a wide range of $c$ values from Corollary 5.3, and thus experimentally check whether theoretical prediction that $\log \mathrm{Bias}^2 / \log \mathrm{Var}$ changes from $0.557$ to $2$. It would be interesting to see whether the theory can work on such a fine level of detail. However, this also contradicts \textit{bias-variance alignment} statement (1) since it requires $\log \mathrm{Bias}^2 / \log \mathrm{Var} \to 1$ as $\log \mathrm{Var}\to-\infty$.

---

> ### Author Response · Authors · 2023-11-21
> **Response to Reviewer ehNn**
>
> Thank you so much for your time, insightful feedback, and the positive evaluation :-)
>
>
> > Q: Formally, eq. (52) is correct only up to the addition of a vector parallel to $\mathbf{1}_K$. Indeed, such vectors, when passed to softmax, can be ignored and therefore does not affect the subsequent calculations performed in the paper. However, it would be better to mention this to avoid confusion.
>
>
> A: Thank you for your careful review and insightful comment. We acknowledge that Eq. (52) is indeed correct only up to the addition of a vector parallel to $\mathbf{1}_K$. As you correctly pointed out, such vectors, when passed to the softmax function, can be ignored and do not affect the subsequent calculations performed in the paper. We appreciate you bringing this to our attention, as it is important to be precise in our mathematical derivations.
>
> In the rebuttal revision, we have added a more detailed, step-by-step derivation of Eq. (52) and emphasized that the addition of a vector parallel to $\mathbf{1}_K$ does not change the value of the softmax function. We believe that this additional explanation will help to clarify the derivation and avoid any further confusion.
>
> > Q: The colors for negative and positive classes in figures 12,13 are swapped compared to the rest of the plots in the papers. Though completely stylistic, it would be nice to have a consistent color scheme to make paper reading more comfortable.
>
> A: Thank you for bringing this to our attention! We acknowledge the inconsistency in the color scheme for figures 12 and 13 compared to the rest of the plots in the paper. We intend to revise these figures to ensure a consistent color scheme throughout the entire presentation. Your observation is much appreciated.
>
> > Q: It took me some time to verify that variable $F$ appearing in eq. (59) indeed has $F(2(K-1),2)$ distribution. Since this fact is central for the following computation and, to my understanding, motivates the choice of Gumbel distribution for $v$ in assumption 5.1, it would be better to discuss and explain this explicitly in appendix.
>
> A: Thank you. In the revision rebuttal, we have added an explanation that this is because $e^{u_i}\sim \mathrm{Exp}(\lambda) = \frac{1}{2\lambda}\chi_2^2$.
>
>
> > Q: Histograms for negative classes have disappeared in the middle row of Figure 12.
>
> A: Thank you for pointing that out. We suspect that the absence of histograms for negative classes in the middle row of Figure 12 might be attributed to a rendering issue with the vector graphics, particularly due to the numerous bins in the negative class histogram. We have opted for vector graphics to enhance the professional appearance of the figure. We are committed to addressing this concern and plan to recreate the figure, aiming to resolve the rendering issue. Your feedback is valuable, and we appreciate your attention to detail.
>
> > Q: A typo "$K$ as the number of samples" at the end of sec. G.2.
>
> A: Thank you! We have fixed this typo.
>
> > Q: Does figure 13 contain only correctly classified samples or all of them? In the latter case, it would be interesting to know why the approximately vertical shape of wrongly classified points (e.g. a "blue tail" on figure 1.b) is not present in the synthetic data.
>
>
>
>
> A: Thank you for your question regarding Figure 13, and we appreciate the opportunity to provide clarification. Figure 13 indeed includes all samples from the dataset, not just the correctly classified ones. The absence of the approximately vertical shape, particularly the "blue tail" observed in Figure 1.b, is attributed to the fact that all the samples in Figure 13 are correctly classified.
>
>
> It's important to note that Figure 13 illustrates the bias-variance alignment on a synthetic dataset generated in accordance with the principles of the neural collapse theory. In this idealized setting, where the data conforms perfectly to the theoretical framework, we observe a full alignment between bias and variance.
>
>
> We appreciate your keen observation, and in response to your question, we have included a sentence in the paper to explicitly state that Figure 13 contains all samples and that they are correctly classified.

---

> > ### Author Response · Authors · 2023-11-21
> > **Response to Reviewer ehNn (continued)**
> >
> > > Q: Choice of Gumbel distribution. Overall, the validation of assumption 5.1 presented in sec. G.2 seems to support the hypothesis that we may write $\psi_\tau(X)=R(sw^{\mathrm{ETF}_Y+v})$ with random vector $v$ whose entries are i.i.d., zero mean, and have unimodal shape. For example, Gaussian distribution satisfies these criteria. So is there some additional evidence for choosing Gumbel distribution, besides it enabling an analytical solution?
> >
> > A:  Indeed, the choice of the Gumbel distribution was primarily motivated by its ability to facilitate an analytical solution, despite the complexity of the mathematical expression. In contrast, opting for a Gaussian distribution introduces challenges, as even the expectation of the softmax of a Gaussian random vector lacks a tractable form. Researchers often have to resort to the mean-field approximation, as illustrated in the following reference:
> >
> > Lu, Z., Ie, E., & Sha, F. (2020). Mean-field approximation to gaussian-softmax integral with application to uncertainty estimation. arXiv preprint arXiv:2006.07584.
> >
> > > Q: After eq. (10) it is mentioned that the paper considers (except sec. E.2) MSE loss on top of classifier probabilities. However, I did not find any mention that this is also applied to the training of neural networks used for empirical results - that would be quite non-standard from a practical point of view (CE loss is usually used in practice). Could you please provide details on how the networks were trained?
> >
> > A: Thank you for your question. We appreciate your attention to detail. We want to clarify that, in practice, all of the neural networks utilized in our study were trained with the Cross-Entropy (CE) loss, adhering to the standard practice in the field. To address this, we have included a clarification at the end of page 3 to explicitly highlight the use of CE loss in the training of our networks. If you have any further questions or require additional details, please feel free to ask.
> >
> > > Q: If, in the previous question, CE loss was used, I would expect that scale parameter $s$ of ETF complex grows with training time in the terminal phase of training (it is required for reduction of train loss to extremely small values). Then, by training the network for a very long time, one can hope to cover a wide range of $c$ values from Corollary 5.3, and thus experimentally check whether theoretical prediction that $\log \mathrm{Bias}^2 / \log \mathrm{Var}$ changes from $0.557$ to $2$. It would be interesting to see whether the theory can work on such a fine level of detail. However, this also contradicts \textit{bias-variance alignment} statement (1) since it requires $\log \mathrm{Bias}^2 / \log \mathrm{Var} \to 1$ as $\log \mathrm{Var}\to-\infty$.
> >
> > A: Thank you for your thoughtful question and insightful observations regarding the evolution of the ETF in the context of neural collapse theory. We appreciate the opportunity to discuss these complex issues with you.
> >
> > The evolution of the ETF with training time, as you rightly pointed out, is an intriguing aspect of our work. We acknowledge that delving into the intricacies of this evolution may necessitate a dedicated exploration in a subsequent paper, accompanied by extensive numerical results to validate theoretical predictions.
> >
> > In our current paper, our primary focus is on examining the alignment between bias and variance as we terminate neural network training. The question of how this alignment evolves with prolonged training is indeed a pertinent one, and we agree that it could be an interesting avenue for future research.
> >
> > Regarding the latter part of your question, we agree with the reviewer that the behavior of the ratio $\log \mathrm{Bias}^2 / \log \mathrm{Var}$ is contingent on the range covered by the parameter $c$ from Corollary 5.3.
> >
> > It is indeed plausible that, with prolonged training, the parameter $c$ might converge to a specific value, resulting in bias-variance alignment. Alternatively, bias-variance alignment could occur when training is halted, possibly when the model has achieved a satisfactory level of performance. We also acknowledge the possibility you raised, where the ratio evolves from $0.557$ to $2, but such a transition might necessitate an impractically long training duration, and practitioners may have terminated training due to the associated increase in validation error.
> >
> > We appreciate your valuable feedback and are open to further discussion on these intriguing aspects of our work. If you have any additional questions or suggestions, please feel free to share them.

---

### Meta-Review · Area_Chair_PnAu · 2023-12-11

**Metareview:**

This paper makes an interesting empirical and theoretical study of the puzzling phenomenon that the bias and variance of per-sample predictions among an ensemble of deep learning models are actually "aligned".

All reviewers thought that this paper makes an interesting contribution to ICLR. The authors have well addressed with their rebuttal and revision the concerns that the reviewers had originally raised. Reviewer iQea increased their score from 5 to 6 after the revision. They still think that some improvements could be made about the related work in the camera ready version (as well about the clarity of the presentation), and the authors are asked to implement these changes for the final version.

**Justification For Why Not Higher Score:**

This paper could also be an oral, depending on the calibration from the SAC.

**Justification For Why Not Lower Score:**

This is a thought provoking paper that could make an interesting spotlight/oral. The bias-variance tradeoff is a commonly held principle by the machine learning community, and so new nuances about it are quite interesting. All reviewers thought this was a strong submission.

---

### Decision · Program_Chairs · 2024-01-16

Accept (spotlight)